# Seeing through Uncertainty: Robust Task-Oriented Optimization in Visual Navigation

**Yiyuan Pan**    **Yunzhe Xu**    **Zhe Liu**\*    **Hesheng Wang**\*
Shanghai Jiao Tong University
{pyy030406, xyz9911, liuzhesjtu, wanghesheng}@sjtu.edu.cn

## Abstract

Visual navigation is a fundamental problem in embodied AI, yet practical deployments demand long-horizon planning capabilities to address multi-objective tasks. A major bottleneck is data scarcity: policies learned from limited data often overfit and fail to generalize OOD. Existing neural network-based agents typically increase architectural complexity that paradoxically become counterproductive in the small-sample regime. This paper introduce NEURO, a integrated learning-to-optimize framework that tightly couples perception networks with downstream task-level robust optimization. Specifically, NEURO addresses core difficulties in this integration: (i) it transforms noisy visual predictions under data scarcity into convex uncertainty sets using Partially Input Convex Neural Networks (PICNNs) with conformal calibration, which directly parameterize the optimization constraints; and (ii) it reformulates planning under partial observability as a robust optimization problem, enabling uncertainty-aware policies that transfer across environments. Extensive experiments on both unordered and sequential multi-object navigation tasks demonstrate that NEURO establishes SoTA performance, particularly in generalization to unseen environments. Our work thus presents a significant advancement for developing robust, generalizable autonomous agents.

**Code:** https://github.com/PyyWill/NeuRO

## 1   Introduction

Visual navigation has emerged as a cornerstone problem in robotics, where agents must reason over complex 3D environments and pursue possible multiple goals under uncertainty. Among benchmark tasks, Multi-Object Navigation (MultiON) task [23] evaluates an agent's ability to locate multiple objects within 3D environments, requiring sophisticated scheduling strategies across multiple goals. This task presents significant challenges due to its multi-goal nature and severe data scarcity, which often causes agents to overfit to training environments and generalize poorly to unseen scenarios. While existing neural network (NN)-based approaches attempt to enhance performance by stacking complex network modules, these methods tend to exacerbate overfitting rather than improve generalization in low-data regimes such as search-rescue missions [19, 24].

A potential solution is instead to couple learning with explicit optimization models to form task-aware training frameworks [9]: *First*, optimization models can explicitly articulate hard task constraints (e.g., collision avoidance in navigation) without additional parameters to capture shared dynamics across related tasks, thus enhancing generalization; *Secondly*, their inherent structure facilitates the simultaneous consideration and scheduling of multiple objectives. Therefore, it's a natural idea

---

\*Corresponding author. The authors are with the School of Automation and Intelligent Sensing, Shanghai Jiao Tong University, and the Key Laboratory of System Control and Information Processing, Ministry of Education of China, Shanghai 200240. Zhe Liu is also with the National Key Laboratory of Human-Machine Hybrid Augmented Intelligence, Institute of Artificial Intelligence and Robotics, Xi'an Jiaotong University.

39th Conference on Neural Information Processing Systems (NeurIPS 2025).

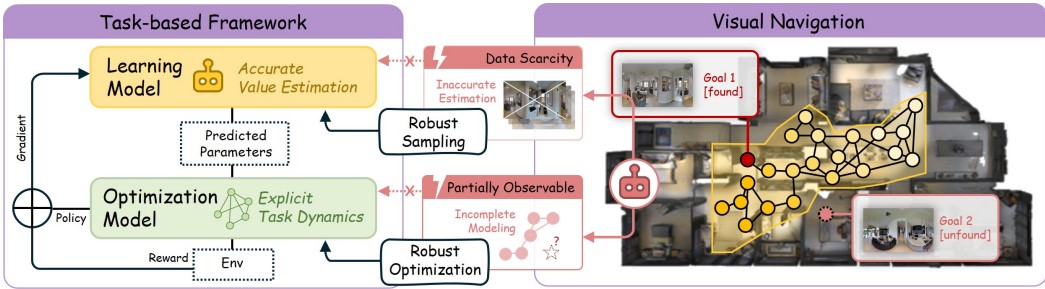

Figure 1: **Overview of the work**. (**Left**) The foundational concept of a task-based optimization framework and its inherent challenges when directly applied to visual navigation. (**Right**) NEURO framework addresses this mismatch through a conformal visual-processing method and a robust optimization formulation for decision-making.

to wed networks with optimization in visual navigation. However, two non-trivial challenges are raised (see Fig. 1): *First*, the reliability of optimization outputs depends critically on the accuracy of parameters predicted by the network and the data scarcity can lead to prediction errors, destabilizing the entire pipeline; *Secondly*, formulating an optimization problem often requires global information, which is unattainable in partially observable environments typical of embodied navigation. In essence, a naïve integration might instead degrade system stability and overall performance.

To this end, we propose NEURO, a robust framework for training visual navigation agents end-to-end with downstream optimization tasks. Specifically, NEURO incorporates two key technical innovations: (i) For the unreliable network predictions, we employ Partially Input Convex Neural Networks (PICNNs) to distill complex, non-convex visual information into high-dimensional convex embeddings. These embeddings are then transformed by a proposed calibration method into a tractable, convex uncertainty set, which is subsequently passed to the downstream optimization problem; (ii) For the issue of partial observability, we formulate the visual navigation (formally modeled as a POMDP) as a pursuit-evasion game, casting it as a robust optimization (RO) problem. Such formulation inherently manages parametric uncertainty arising from partial observations and demonstrates generality across diverse navigation tasks. We evaluate NEURO on our proposed unordered MultiON and traditional sequential MultiON tasks, demonstrating superior performance and generalization over state-of-the-art (SoTA) network-based approaches. In summary, our contributions are threefold:

- We propose NEURO, a novel hybrid framework that synergistically integrates deep neural networks with downstream optimization tasks for end-to-end training, significantly improved generalization in data-scarce regimes.

- We introduce a methodology to bridge unreliable navigation prediction and safe trajectory optimization. By leveraging PICNN-based conformal calibration and casting POMDP planning as robust optimization, NEURO effectively handles partial observability and parametric uncertainty.

- Extensive empirical validation demonstrating that NEURO establishes superior performance on challenging MultiON benchmarks, significantly outperforming existing methods, particularly in generalization to unseen environments.

## 2 Related Works

**Visual Navigation.** Visual navigation [6, 7, 13, 4, 20] are cornerstone research areas in embodied AI. A particularly challenging sub-domain is Multi-Target Object Navigation (MultiON) [23], which demands sophisticated, long-horizon decision-making for locating multiple objects. Prevailing approaches in MultiON and broader visual navigation often augment agents via enhanced memory [15, 10] or predictive world models [21, 17]. While aiming for comprehensive environmental understanding [10, 17, 21, 15], this pursuit frequently results in data-hungry models and excessive training overhead, without directly optimizing the crucial decision-making process itself [5, 8]. Such "black-box" decision mechanisms, often reliant on complex neural networks navigating the

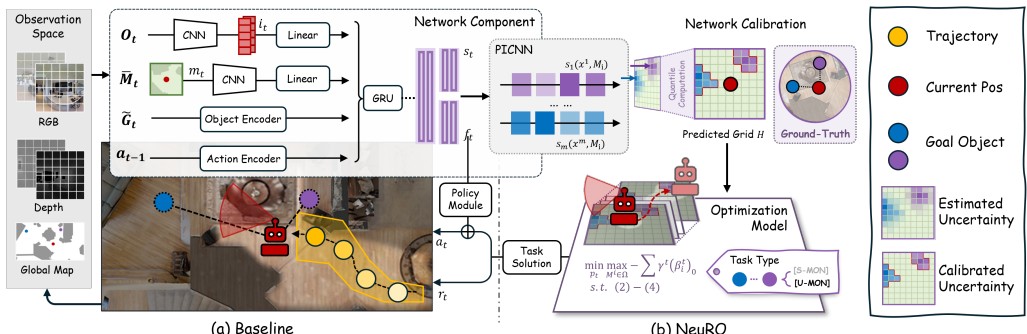

Figure 2: **Architecture of NEURO**. (**Left**) The decision-making process of a purely network-based agent at each navigation step. (**Right**) Our introduced optimization module, which redirects the agent's training back toward the task itself.

exploration-exploitation trade-off [22, 16], can suffer from information loss and high training complexity, especially under data scarcity, hindering robust and interpretable actions. Recognizing these limitations, particularly for demanding tasks like MultiON, our work departs from purely representational learning and advances a task-based training paradigm [1]. We advocate for integrating explicit optimization models within the decision pipeline, thereby refocusing learning towards task-specific optimal behaviors and improving decision efficiency.

**Task-Based Optimization.** The concept of "task-based" end-to-end model learning was introduced by [9], which proposed training machine learning models in a way that captures a downstream stochastic optimization task. Later, the applicability of this framework was extended to various types of convex optimization problems via the implicit function theory [1, 2]. Subsequently, more sophisticated learning models were investigated in the context of the task-based model to accommodate more complex task needs [12, 26, 14, 25]. Nevertheless, prior studies have primarily focused on short-horizon tasks such as portfolio management or inventory control, typically assuming access to complete and well-structured datasets. In contrast, embodied robotic tasks often operate in data-scarce settings and require long-horizon planning under uncertainty. Building on this gap, we propose NEURO, the first task-based learning framework explicitly designed for embodied agents' tasks.

## 3 Method

**Problem Setup.** We formalize MultiON for both unordered (U-MON) and sequential (S-MON) object discovery. In each episode, an agent navigates a 3D environment to locate a sequence of $m$ target objects selected from a candidate set $G$ of size $k$. At each time step $t$, the agent receives an egocentric observation comprising RGB-D images $o_t$ and a $k$-dimensional binary vector $\tilde{G}_t$. For U-MON, $\tilde{G}_t$ indicates the $n$ ($n \leq m$) remaining targets with $n$ entries set to 1; for S-MON, $\tilde{G}_t$ identifies the single current target (i.e., $n = 1$). The agent selects an action $a_t$ from the set {FORWARD, TURN-LEFT, TURN-RIGHT, FOUND} based on $o_t$, $\tilde{G}_t$, and its internal state. Upon executing a FOUND action, if a valid target is within a predefined proximity, $\tilde{G}_t$ is updated. An episode succeeds if all $m$ targets are found; it fails if the maximum step limit is reached or an incorrect FOUND action is performed.

**Method Overview.** Fig. 2 illustrates the operational pipeline of NEURO. At its core, this work addresses the fundamental challenge of bridging neural networks with optimization models. Building upon the foundational network components (Section 3.1) and the optimization formulation (Section 3.2), we then elucidates how NEURO tackles two pivotal questions: (i) *How does NEURO transform network predictions into reliable inputs for optimization? (Section 3.3)* (ii) *How does NEURO exploit optimization feedback to improve network training? (Section 3.4)*

### 3.1 Neural Perception Module

At each step $t$, the agent processes its egocentric RGB image $c_t$, depth image $d_t$ from observation $o_t = (c_t, d_t)$, and maintains an egocentric map view $m_t$, which is a partially observed, agent-centric

perspective (rotated and cropped) of an inaccessible underlying global map $\bar{M}$. A CNN block extracts visual features $i_t$ from $o_t$, which are then linearly embedded into a vector $v_i$. Similarly, the egocentric map $m_t$ is embedded into $v_m$. These two embeddings are concatenated with a goal object embedding $v_g$ (from $\tilde{G}_t$) and an action embedding $v_a$. The resulting tensor $v_t = \text{concat}(v_i, v_m, v_g, v_a)$ is fed into a GRU to compute the hidden state $s_t$ and a state feature $f_t = \{f_t^i\}_{i=1}^n$, corresponding to the $n$ pertinent objects.

Next, from the processed features $f_t$, a policy module parameterized by $\theta$ derives a preliminary action policy $\pi(a_t^{\text{net}}|f_t)$ and an approximate value function $V_\theta(s_t)$. During training, the agents will receive an environmental reward signal $r_t^{\text{env}}$ according to its chosen action $a_t$, which encourages goal-reaching efficiency while penalizing delays. For traditional purely network-based agents (our baseline), the final actions $a_t$ are equivalent to these network-generated actions, $a_t \equiv a_t^{\text{net}}$. However, the efficacy of such an approach is critically contingent upon the fidelity of $f_t$, which often necessitates a comprehensive world understanding. Thus, such agents can impose a significant training burden and increase susceptibility to overfitting under data-scarce conditions.

### 3.2    Robust Optimization Planner

We model the MultiON task as a robust pursuit-evasion problem to handle the inherent partial observability and uncertainty in target locations. In this formulation, the agent and $n$ remaining target objects are situated on a discrete grid $H$ with $E$ edges and $V = E \times E$ cells, which serves as an abstraction to simulate the agent's limited local field of view. Due to the partial observability, the agent doesn't know the exact objects' positions but instead maintains a belief about their potential movement, represented by a set of transition matrices $\{M^i\}_{i=1}^n \in \Omega$ (we omit episode's timestep $t$). $M_{uv}^i$ denotes the agent's estimated probability that object $i$ transitions from cell $u$ to cell $v$ in one time step. As navigation unfolds, the agent belief evolves by refining likely objects locations. These estimated matrices are the source of uncertainty that our robust optimization will address.

We first define the agent positions $p_t$ as a continuous variable for gradient-based optimization, while the objects' locations are mapped to these discrete cells. Then, the legality of the agent's path is ensured with the following constraints, where $\bar{d}$ is the maximum travel distance.

$$||p_{t+1} - p_t||_2 \leq \bar{d}, \quad \forall t \in \{0, \ldots, \tau - 1\} \tag{1a}$$

$$p_0 = [E/2 \quad E/2]^T \tag{1b}$$

$$0 \leq (p_t)_x, (p_t)_y \leq E, \quad \forall t \in T \setminus \{0\} \tag{1c}$$

Next, we introduce two key variables to track the evolving understanding of target locations: (i) *Prior Belief State* $\beta_i^t$: A $(V + 1)$-dimensional vector representing the agent's confidence regarding object $i$'s presence in each of the $V$ grid cells and its capture status. Specifically, $(\beta_i^t)_v$ for $v \in \{1, \ldots, V\}$ quantifies the belief that object $i$ is in cell $v$, and a value of 0 implies the agent believes the cell is empty of object $i$; the 0-th entry, $(\beta_i^t)_0$, represents a *global capture belief*, where a value approaching 1 signifies high confidence in successful capture, see Eq. (2c). By definition, the total belief sums to one, as successful capture implies the elimination of uncertainty across all grid cells for that object. The initial *Prior Belief State*, $\beta_i^0$, typically reflects no prior information, assigning a uniform belief of all cells except the agent's starting cell, see Eq. (2d). (ii) *Posterior Belief State* $\alpha_i^t$: A $V$-dimensional vector serving as an intermediate representation. It is derived by applying the agent's objects transition matrix $M^i$ to the non-captured components of the *Prior Belief State* $\beta_i^{t-1}$, as shown in Eq. (2b). $\alpha_i^t$ represents the belief distribution over grid cells after accounting for potential object "movement" but before incorporating new observations from time $t$.

$$\alpha_i^t, \beta_i^t \in [0, 1], \forall t \in T \tag{2a}$$

$$(\alpha_i^t)_v = \sum_{u \in V} M_{uv}^i (\beta_i^{t-1})_u, \forall t \in T \setminus \{0\}, v \in V \tag{2b}$$

$$(\beta_i^t)_0 = 1 - \sum_{v \in V} (\beta_i^t)_v, \ \forall t \in T \setminus \{0\}, \ v \in V \tag{2c}$$

$$(\beta_i^0)_v = c = \begin{cases} 0, & \text{if } v = 0 \text{ or } v = \left\lfloor \frac{V}{2} \right\rfloor + 1 \\ \frac{1}{V-1}, & \text{otherwise} \end{cases} \tag{2d}$$

The capture determination employs a *same-cell binary detection* model: if the agent occupies cell $v$ at time $t$, it can ascertain object $i$'s presence in cell $v$ and eliminate local uncertainty. Consequently,

the updated *Prior Belief State* $(\beta_i^t)_v$ for $v \in V$ can be expressed as the element-wise product of the *Posterior Belief State* $\alpha_i^t$ and an *Detection Uncertainty Factor* $d(p_t)_v$, as shown in Eq. (3a). The detection uncertainty $d(p_t)_v$, see Eq. (3c), is a function of the agent's current position $p_t$ and the cell index $v$. Intuitively, for cells distant from the agent $(d(p_t)_v \to 1)$, the posterior belief $(\alpha_i^t)_v$ equals to the prior belief $(\beta_i^t)_v$, as no new reliable information is obtained; for cells at or very near the agent's position $(d(p_t)_v \to 0)$, uncertainty is significantly reduced as $(\beta_i^t)_v$ approaches 0.

$$(\beta_i^t)_v = (\alpha_i^t)_v d(p)_v, \ \forall t \in T \setminus \{0\}, \ v \in V \tag{3a}$$

$$d(p_t)_v = \frac{|v_x - (p_t)_x| + |v_y - (p_t)_y|}{V - 1}, \ \forall t \in T \tag{3b}$$

$$v_x = (v \bmod E) + 0.5, \ v_y = \lfloor v/E \rfloor + 0.5 \tag{3c}$$

Finally, we present the objective function of the optimization problem. For the U-MONtask, the objective is to find an agent path $p_0, \ldots, p_\tau$ that maximizes the accumulated discounted capture belief, robust to the uncertainty in $M^i$. The formulation for S-MONis provided in the Appendix A.

$$[\text{U-MON}] \min_{p_t} \max_{M^i \in \Omega} - \sum_{t \in T} \gamma^t \sum_{i \in n} (\beta_i^t)_0 \tag{4}$$

$$s.t. \quad (2) - (4)$$

## 3.3 Learning-Planning Interface: Prediction Calibration

This section details how NEURO transforms raw network outputs $x_t^i = \text{concat}(f_t^i, s_t)$ for each object into a well-defined objects transition matrix $M_t^i$ under uncertainty. We omit the object index $i$ and time index $t$ or conciseness. We first give a universal formulation of uncertainty mapping with a general nonconformity score function $g(x, M)$:

$$\Omega(x) = \{M \in \mathbb{R}^V \mid g(x, M) \leq q\} \tag{5}$$

The selection of the nonconformity score function $g(x, M)$ is critical. To ensure that the uncertainty set $\Omega(x)$ is convex in $M$—a property pivotal for tractable downstream optimization—we employ PICNNs [3] to instantiate $g(x, M)$ (details in Appendix B), yielding a $g(x, M)$ convex in $M$ for any fixed $x$. Furthermore, PICNNs can approximate complex, non-convex visual landscapes by learning a collection of conditional convex functions, offering a structured and expressive uncertainty representation from rich visual inputs.

With $g(x, M)$ established through a PICNN, defining the final uncertainty set $\Omega(x)$ then hinges on the principled selection of the threshold $q$, as it balances the risk of an overly optimistic set against an uninformatively large one. $\Omega(x)$ provides a statistically sound basis for robust decision-making, we desire $q$ to reliably captures the underlying true object transition matrix with a prespecified coverage level $\alpha$, as formalized by the Definition 1 below.

**Definition 1 (Marginal Coverage)** *An uncertainty set generator $\Omega(\cdot)$ provides marginal coverage at the $(1 - \alpha)$-level for an unknown data distribution $\mathcal{P}$ if $\mathbb{P}_{(x,M) \sim \mathcal{P}}(M \in \Omega(x)) \geq 1 - \alpha$.*

Given the definition, the relationship between the threshold $q$ and the desired coverage level $\alpha$ can be established by the following proposition (proof in Appendix C). Accordingly, we compute $q$ by sorting the scores $\{g(x_t^i, M_t^i)\}_{n=1}^N$, obtained from an i.i.d. calibration set sampled at time $t$ for object $i$, in ascending order and selecting the score at the $\lceil (N_{calib} + 1)(1 - \alpha) \rceil$-th position.

**Proposition 1 (Coverage with Quantile)** *Let the dataset $\mathcal{D} = \{(x_n, M_n)\}_{n=1}^N$ to be sampled i.i.d from the implicit distribution $\mathcal{P}$ gained during training phase. And $q$ is set to be the $(1 - \alpha)$-quantile for the set $\{g(x_n, M_n)\}_{n=1}^N$, then $\Omega(x)$ gains the following guarantee.*

$$1 - \alpha \leq \mathbb{P}_{x, M, \mathcal{D}}(M \in \Omega(x)) \leq 1 - \alpha + \frac{1}{N + 1}$$

Although the uncertainty set $\Omega(x)$ is now well-defined, the original RO problem in Eq. (4) remains intractable directly due to its inherent min-max structure. To overcome the issue, we reformulate it as a solvable, linear counterparts by taking the dual of the inner maximization problem and invoking

strong duality (see detailed derivation and S-MON*'s formulation in Appendix D):

$$[\text{U-MON}^*] \min_{p,\pi,\mu,\lambda,\nu,\xi,k} \sum_{i\in n, t\in T, v\in V} -c^T\mu_i - \lambda_i^t - (b_i^v)^T\xi_i^v$$

$$s.t. \quad (2) - (4)$$
$$D_t^T\nu_i^0 \le 0, \ \forall i \in n$$
$$D_t^T\nu_i^t - \sum_{v\in V}\pi_{i,v}^t T_v^T \le 0, \forall i \in n, \ t \in T \setminus \{0\}$$
$$\sum_{t\in T}\pi_{i,v}^t C\beta_i^t - (A_i^v)^T\xi_i^v \le 0, \xi_i^v \ge 0, \ \forall i \in n, v \in V \tag{6}$$
$$(\Gamma^t)^T + (\nu_i^t)^T S + \lambda_i^t E + G_{i,t} \ge 0, \forall i \in n, \ t \in T$$
$$G_{i,t} = \begin{cases} \mu_i^T - \sum_{v\in V}\pi_{i,v}^0(k_i^v)^T C, & \text{if } t=0 \\ 0, & \text{if } t=\tau \\ -\sum_{v\in V}\pi_{i,v}^t(k_i^v)^T C, & \text{otherwise} \end{cases}$$

where $C, S, T_v, E, \Gamma^t$ are constant value matrices and vectors. $\pi_{i,v}^t, \lambda_i^t \in \mathbb{R}$, $\mu_i \in \mathbb{R}^{V+1}$, $\nu_i^t \in \mathbb{R}^V$, $\xi_i^v \in \mathbb{R}^{2Ld+1}$, $k_i^v \in \mathbb{R}^{V+Ld}$ are decision variables. $D_t \in \mathbb{R}^{V\times V}$ is the diagonal matrix of vector $d(p_t)$. $b \in \mathbb{R}^{2Ld+1}$, $A \in \mathbb{R}^{(2Ld+1)\times(V+Ld)}$ are constructed from the PICNN's inherent parameters.

### 3.4 Planning-Learning Interface: Solution Feedback

To enable end-to-end training of our hybrid NEURO agent, the output of the downstream optimization problem is integrated back into the neural network module in two primary ways. *First*, a refined action signal is derived. The optimal trajectory $p^* = \{p_t^*\}_{t=0}^\tau$ from the optimization solution provides a goal-directed action offset, denoted $a_t^{\text{task}}$. This offset is then combined with the network's original action proposal $a_t^{\text{net}}$. *Secondly*, the optimal objective value of the optimization problem serves as a task-specific reward signal, $r_t^{\text{task}}$, which is used to augment the agent's intrinsic reward $r_t^{\text{env}}$. The gradient of this optimization-derived reward $r_t^{\text{task}}$ with respect to the network parameters $\theta$ can be computed via the chain rule, leveraging implicit differentiation through the Karush-Kuhn-Tucker (KKT) conditions. These integrations are formalized as:

$$\begin{cases} a_t \triangleq \lambda \cdot a_t^{\text{net}} + (1-\lambda) \cdot a_t^{\text{task}} \\ r_t \triangleq \text{GVM}(r_t^{\text{env}}, r_t^{\text{task}}) \end{cases} \tag{7a}$$

$$\partial r_t^{\text{task}}/\partial\theta = \partial r_t^{\text{task}}/\partial p^* \cdot \partial p^*/\partial\theta \tag{7b}$$

where $\lambda$ is a blending factor for actions and GVM denotes Goal Vector Method, a parameter-free approach that combines multiple reward signals to guide the policy towards Pareto-optimal solutions, mitigating issues of conflicting gradient directions from naïvely summed rewards.

Furthermore, we provide an additional optimization-based theoretical guarantee regarding optimality and convergence in Appendix E to ensure the stability of this learning process.

---

**Algorithm 1** NeuRO Training Algorithm

1: **initialize** episode.
2: **for** $t$ in $max\_steps$ **do**
3:    **receive** $o_t$ and $\tilde{G}_t$ from $env \Rightarrow (s_t, f_t)$
4:    **output:** $\Omega(\cdot)$   *// uncertainty calibration*
5:    **solve** $[\text{U-MON}]^*$ or $[\text{S-MON}]^*$.
6:    **apply** offset $a_t^{\text{task}}$ and $r_t^{\text{task}}$
7:    **receive** $S_{t+1}$ and reward $r_t$
8:    **update** $\theta$ with $\nabla_\theta r_t$
9:    **if** $\tilde{G}_t = 0$ **then**
10:      **break**
11:    **end if**
12: **end for**

---

## 4  Experiments

**Tasks and Evaluation Metrics.** We test on unseen settings with object counts $m = 1, 2, 3$ across standard train, validation and test splits from U-MON and S-MON tasks. We adopt four commonly used evaluation metrics from visual navigation studies. 1) Success: A binary indicator of episode success; 2) Progress: The fraction of object goals successfully FOUND; 3) SPL (success weighted by path length): Total traveled distance weighted by the sum of the geodesic shortest path from the agent's starting point to all the goal positions; 4) PPL: A variant of SPL that weights based on progress.

Table 1: Comparison with existing methods on U-MON$_{m=2,3}$ and S-MON$_{m=2,3}$ tasks.

| Methods | S-MON$_{m=2}$: Test | | | | S-MON$_{m=3}$: Test | | | |
|---|---|---|---|---|---|---|---|---|
| | Success↑ | Progress↑ | SPL↑ | PPL↑ | Success↑ | Progress↑ | SPL↑ | PPL↑ |
| SMT[11] | 28 | 44 | 26 | 36 | 9 | 22 | 7 | 18 |
| FRMQN[18] | 29 | 42 | 24 | 33 | 13 | 29 | 11 | 24 |
| OracleMap (Occ) | 34 | 47 | 25 | 35 | 16 | 36 | 12 | 27 |
| ProjNeuralMap[23] | 45 | 57 | 30 | 39 | 27 | 46 | 18 | 31 |
| ObjRecogMap[23] | 51 | 62 | 38 | 45 | 22 | 40 | 17 | 30 |
| OracleEgoMap | 64 | 71 | 49 | 54 | 40 | 54 | 25 | 36 |
| OracleMap | 74 | 79 | 59 | 63 | 48 | 62 | 38 | 49 |
| Lyon[15] | 76 | 84 | 62 | 70 | 57 | 70 | 36 | 45 |
| HTP-GCN[17] | 76 | 84 | 60 | 67 | 57 | 70 | 27 | 33 |
| **NEURO (Ours)** | **80** | **86** | **66** | **72** | **62** | **72** | **40** | **47** |

| Methods | U-MON$_{m=2}$: Test | | | | U-MON$_{m=3}$: Test | | | |
|---|---|---|---|---|---|---|---|---|
| | Success↑ | Progress↑ | SPL↑ | PPL↑ | Success↑ | Progress↑ | SPL↑ | PPL↑ |
| OracleEgoMap | 54 | 62 | 38 | 44 | 36 | 42 | 33 | 38 |
| **NEURO (Ours)** | **62** | **66** | **52** | **56** | **45** | **53** | **45** | **50** |

## 4.1 Comparison with SoTA

Table 1 compares NEURO's performance against other agents on the S-MONtask ($m = 2, 3$), using OracleEgoMap (visualized in Fig. 2) as our network-based baseline. Our NEURO framework empowers the agent to achieve superior performance on the S-MONbenchmark. Notably, NEURO demonstrates a substantial $4\%$ improvement in SPL (without data augmentation), indicating enhanced navigation efficiency. This efficiency gain is attributed to the agent's ability to leverage the downstream optimization problem for more effective global path planning. Meanwhile, the performance gain is more pronounced at $m=3$ compared to $m=2$, which we attribute to the optimization model's ability to coordinate multiple targets. Furthermore, the NEURO architecture exhibits remarkable adaptability stemming from the customizable nature of its optimization component. By solely modifying the downstream optimization model—without requiring retraining of networks—the agent can efficiently adapt to distinct task formulations (i.e., S-MON and U-MON).

## 4.2 Quantitative and Qualitative Study

**Quantitative Study.** As illustrated in Fig. 3, the NEURO training framework facilitates significantly accelerated convergence with respect to the Success rate on specific tasks, ultimately surpassing the performance benchmarks set by purely network-based methodologies. This rapid learning curve strongly suggests an enhanced capability of NEURO agents to achieve superior task outcomes, particularly under conditions of data scarcity. Consequently, the NEURO architecture demonstrates heightened sample efficiency and a diminished requirement for extensive training datasets, underscoring its adaptability and practical utility in resource-constrained settings.

**Qualitative Study.** Our NEURO agent demonstrates the capability to generate straightforward yet effective internal representations that aid its decision-making process in visual navigation, as illustrated in Fig. 4. Specifically, we examine the object transition matrix $M_i^t$ for an object $i$, which is derived from the solution of the downstream optimization model. The visualization reveals distinct behaviors of $M_i^t$: for objects currently outside the agent's field of view, $M_i^t$ tends to reflect a diffuse or less certain belief state over possible locations. Conversely, when an object is clearly observed, the corresponding $M_i^t$ sharply localizes, accurately reflecting the agent's high confidence in the object's position on the internal grid $H$.

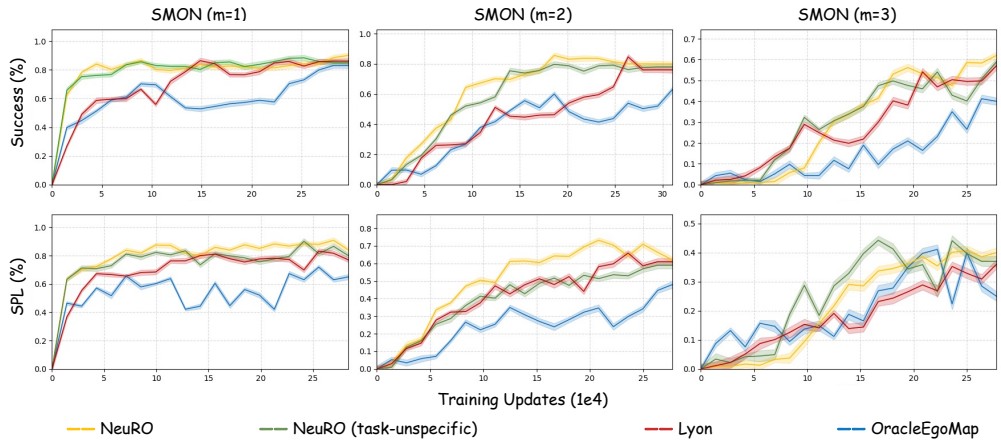

Figure 3: Learning curves for NEURO and baseline during training for different tasks.

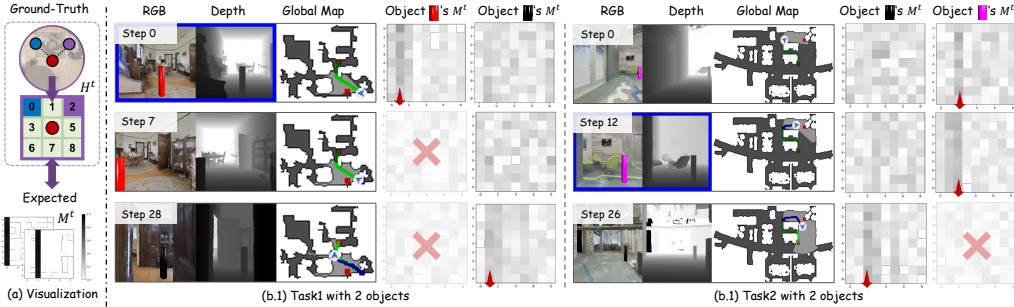

Figure 4: **Visualization of the learned object transition matrix** $\mathrm{M}_i^t$. (**Left**) Mapping objects to optimization grid $H$ and their belief representation in matrix $M_i^t$; darkened columns in $M$ denote high-confidence presence in corresponding cells. (**Right**) navigation scenarios, with the red arrows point to the cell $v$ where the agent predicts the target object is most likely located.

It is crucial to note that this discriminative belief representation is learned without direct supervised labels for $M_i^t$. Its emergence is attributed to the implicit feedback loop established by the optimization component, guiding the network to produce features conducive to effective planning. This underscores the efficacy of leveraging an integrated optimization model to distill actionable insights from uncertainty and enhance the learning outcomes of navigation agents.

Table 2: Impact of various coverage levels

| # | $\alpha$ | Task Performance | | Prediction Error | |
|---|---|---|---|---|---|
| | | Success | SPL | Mean | Variance |
| **1** | 0.01 | 80 | 66 | 0.90 | 0.015 |
| **2** | 0.05 | 76 | 61 | 0.84 | 0.023 |
| **3** | 0.10 | 72 | 56 | 0.80 | 0.047 |
| **4** | 0.20 | 66 | 52 | 0.76 | 0.062 |

### 4.3 Ablation Studies

**Conformal Coverage Level $(1 - \alpha)$.** Table 2 presents an ablation study on the desired marginal coverage level $(1 - \alpha)$ for the conformal prediction sets, examining its impact on both task performance and the agent's predictive accuracy in S-MON$(m = 2)$. As observed, decreasing the target coverage results in less conservative uncertainty sets $\Omega(x)$, which consequently leads to poorer performance as agents fail to make robust estimations and decisions.

To further assess the agent's predictive accuracy (termed 'Prediction Error' in Table3), we evaluated the precision of the learned $M_i^t$. For each object $i$, we derived a belief distribution over the $V$ grid cells from $M_i^t$. Non-Maximum Suppression (NMS) was then applied to identify the cell with the highest belief. This predicted cell index was compared against the ground-truth object cell (discretized based on its relative angle to the agent), and the prediction error was calculated. The mean and

Table 3: Impact of task weight $\lambda$ (**Left**) and optimization scale $(E, \tau)$ (**Right**): S-MON$_{m=2}$.

| | | **Task Weight $\lambda$** | | | | **Optimization Scale $(E, \tau)$** | | | | |
| # | $\lambda$ | Success↑ | Progress↑ | SPL↑ | PPL↑ | $E$ | $\tau$ | Success | Progress | SPL | Inference Time (s) |
|---|---|---|---|---|---|---|---|---|---|---|---|
| 1 | 0.05 | 67 | 74 | 50 | 53 | 3 | 4 | 75 | 79 | 64 | 0.008 |
| 2 | 0.1 | 77 | 81 | 59 | 64 | 3 | 6 | 76 | 80 | 66 | 0.040 |
| 3 | 0.2 | 80 | 86 | 66 | 72 | 5 | 4 | 80 | 86 | 69 | 0.547 |
| 4 | 0.4 | 52 | 57 | 35 | 40 | 5 | 6 | 80 | 85 | 72 | 1.084 |

standard deviation of this prediction error are reported. The results show that as $\alpha$ increases, the mean prediction error also tends to increase, suggesting object location estimates derived from them become less reliable. This corroborates the observed decline in navigation performance, as a less accurate understanding of target locations naturally hampers efficient navigation.

**Action Blending Factor $\lambda$.** Table 3 (Left) displays the impact of varying the action blending factor $\lambda$ on the agent's final task performance in S-MON($m = 2$). We observed that initially, increasing the weight assigned to $a_t^{\text{task}}$ correlates with improved task performance. However, beyond a certain threshold of 0.4, an excessive reliance on $a_t^{\text{task}}$ leads to a notable degradation in navigation performance (approx. 20% drop in Success). This decline is likely because a diminished role for the network's exploratory actions and insufficient exposure to environmental reward signals $r_t^{\text{env}}$ hinder the agent's ability to learn an accurate internal world model. Consequently, the learned object transition matrix $M_i^t$ may fail to capture meaningful environmental dynamics, appearing less informative for effective long-term planning.

**Spatial and Temporal Tuning.** We also investigate the influence of the optimization problem's scale—specifically, the maximum number of optimization steps $\tau$ and the size of the optimization grid $H$—on the agent's navigation performance, with results presented in Table 3 (Right). These parameters collectively define the agent's spatiotemporal planning horizon. Our findings indicate that an increase in the scale of the optimization problem generally correlates with an improvement in navigation Success. Concurrently, while the inference time for solving the optimization problem does increase with scale, it remains consistently low overall. This highlights a key advantage of our parameter-free optimization component, which avoids the computational overhead typically associated with training additional complex predictive modules.

To better approximate real-world complexities and demonstrate the scalability, we further explored larger grid sizes and developed an acceleration technique based on basis function expansion and sparse kernel approximation to maintain computational tractability; however, to maintain focus on NEURO framework in the main paper, these extensions are deferred to Appendix F.

**Task-Specific Optimization.** Finally, we investigated the benefits of task-specific learning on S-MON$_{m=2}$ task, with results in Table 4. To this end, we compared the agent's performance on the S-MON task under two optimization formulations: (i) The full S-MON$^*$ model, which incorporates constraints specifically designed

Table 4: Impact of task-tailored optimization

| # | **Formulation** | Success | Progress | SPL |
|---|---|---|---|---|
| 1 | [S-MON]$^*$ | 80 | 86 | 66 |
| 2 | [S-MON]$^\#$ | 76 | 82 | 60 |

for sequential target acquisition. (ii) A simplified formulation, denoted S-MON$^\#$, which is structurally equivalent to the U-MON$^*$ model (with $n = 1$ target). Notably, there is a 6% improvement in SPL, indicating that the explicit inclusion of task-specific objectives effectively guides the agent towards more efficient, shorter trajectories. This finding validates our approach of adapting the optimization component to better align with the nuances of different visual navigation tasks.

**Generalization Across Task Variations.** To assess the generalization capabilities of our agents, we conducted experiments where models trained on a specific instance of the S-MONtask, denoted S-MON$_{m=i}$ (where $m$ represents the number of navigation goals, and $i$ is the specific number of sub-goals used for training), were evaluated on different instances SMON$_{m=j}$. The performance, typically measured by success rate, is reported in Table 5.

The results in Table 5 indicate a common trend: agents trained on tasks with fewer navigation goals (e.g., $m = 1$) tend to struggle when generalizing to tasks requiring a larger number of goals (e.g.,

Table 5: Success scores on S-MONtasks. Agents are trained on task S-MON$_{m=i}$ (rows) and evaluated on task S-MON$_{m=j}$ (columns). The table below indicates the relative score drop (%) compared to in-task learning performances (diagonal).

| $m$: train \eval | OracleEgoMap | | | Lyon (SoTA) | | | NEURO | | |
|---|---|---|---|---|---|---|---|---|---|
| | **1** | **2** | **3** | **1** | **2** | **3** | **1** | **2** | **3** |
| 1 | 83 | 47 | 28 | 86 | 61 | 50 | 90 | 68 | 57 |
| 2 | 79 | 64 | 32 | 82 | 76 | 54 | 88 | 80 | 60 |
| 3 | 77 | 63 | 37 | 81 | 75 | 57 | 88 | 79 | 62 |
| 1 | 0 | -17 | -9 | 0 | -15 | -7 | 0 | **-12** | **-5** |
| 2 | -4 | 0 | -5 | -4 | 0 | -3 | **-2** | 0 | **-2** |
| 3 | -6 | -1 | 0 | -5 | -1 | 0 | **-2** | **-1** | 0 |

$m = 3$). However, the NEURO framework appears to alleviate this degradation. The average drop in performance when transferring agents trained on a specific $m$ to other $m$ values is comparatively lower for NEURO. We attribute this improved generalization to the embedded optimization model, which likely captures underlying task structures and common rules that are transferable across variations in the number of goals. This allows NEURO to adapt more robustly to related but unseen task configurations.

## 5 Conclusion

We introduced NEURO, a pioneering framework enabling, for the first time, end-to-end training of neural networks with downstream robust optimization for visual navigation. NEURO tackles the critical challenge of agent generalization in data-scarce, partially observable environments by synergistically integrating deep learning's perceptual strengths with robust optimization's principled, uncertainty-aware decision-making. Extensive experiments on U-MON and S-MON benchmarks demonstrate NEURO's superior performance and adaptability. This work not only establishes a promising new paradigm for developing effective and robust embodied AI agents for navigation but also highlights that NEURO's core tenet—fusing learned predictions with formal optimization—extends well beyond this domain (e.g., power dispatch, see Appendix G), moving an important step towards more robust and capable AI systems. Such advancements have broad societal implications, and we acknowledge the need for consideration of ethical implications, such as ensuring safe deployment and mitigating potential biases in learned behaviors.

**Limitations and Future Work.** A key limitation is NEURO's reliance on convex approximations for uncertainty. While PICNNs effectively generate tractable convex uncertainty sets—a significant step—they cannot fully capture inherently non-convex uncertainties common in complex visual scenarios. Addressing general non-convex robust optimization remains a formidable open challenge. Our use of PICNNs is thus a pragmatic, effective, yet approximate strategy. Future work will explore advanced techniques to model and incorporate non-convex uncertainty, aiming to further enhance agent robustness in complex real-world settings.

## Acknowledgment

This paper was supported by the Natural Science Foundation of China under Grant 62303307, 62225309, U24A20278, 62361166632, U21A20480, and Sponsored by the Oceanic Interdisciplinary Program of Shanghai Jiao Tong University, and in part by National Key Laboratory of Human Machine Hybrid Augmented Intelligence, Xi'an Jiaotong University (No. HMHAI-202408).

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

# A Formulation of S-MON

We now present the optimization formulation of S-MON. Unlike U-MON, S-MON does not require balancing between multiple objectives. Therefore, in this setting, we additionally aim to reach the goal via the fastest possible path, rather than just prioritizing information (belief) gain.

$$[\text{S-MON}] \min_{p_t} \max_{M^i \in \Omega} - \sum_{t \in T} \gamma^t \sum_{i \in n} [(\beta_i^t)_0 + \Delta p_t]$$

$$s.t. \quad ||p_{t+1} - p_t||_2 \leq \bar{d}, \quad \forall t \in \{0, \ldots, \tau - 1\}$$

$$p_0 = [E/2 \quad E/2]^T$$

$$0 \leq (p_t)_x, (p_t)_y \leq E, \quad \forall t \in T \setminus \{0\}$$

$$\alpha_i^t, \beta_i^t \in [0, 1], \forall t \in T$$

$$(\alpha_i^t)_v = \sum_{u \in V} M_{uv}^i (\beta_i^{t-1})_u, \forall t \in T \setminus \{0\}, v \in V$$

$$(\beta_i^t)_0 = 1 - \sum_{v \in V} (\beta_i^t)_v, \quad \forall t \in T \setminus \{0\}, \ v \in V$$

$$(\beta_i^0)_v = c = \begin{cases} 0, & \text{if } v = 0 \text{ or } v = \lfloor \frac{V}{2} \rfloor + 1 \\ \frac{1}{V-1}, & \text{otherwise} \end{cases}$$

$$(\beta_i^t)_v = (\alpha_i^t)_v d(p)_v, \quad \forall t \in T \setminus \{0\}, \ v \in V$$

$$d(p_t)_v = \frac{|v_x - (p_t)_x| + |v_y - (p_t)_y|}{V - 1}, \quad \forall t \in T$$

$$v_x = (v \bmod E) + 0.5, \ v_y = \lfloor v/E \rfloor + 0.5$$

$$\Delta p_t = ||p_{t+1} - p_t||_2, \quad \forall t \in \{0, \ldots, \tau - 1\}$$

# B Architecture of PICNNs

To model the uncertainty set in a flexible and general manner, we adopt the PICNN as the score function $g$. Compared to traditional box or ellipsoidal uncertainty sets, which impose shape priors and thus arbitrarily restrict the form of the uncertainty, the PICNN can approximate the true uncertainty landscape—often non-convex—by composing multiple convex components, providing a more expressive and general representation. Using PICNNs allows us to represent complex uncertainty sets while preserving convexity with respect to a subset of the inputs. For simplicity of notation, we omit the index $i$. Specifically, since the PICNN architecture ensures convexity only with respect to its second input, we define $\Omega(x)$ as the sub-level set of $g(x, M)$ with $x$ fixed and $M$ varying, i.e.,

$$\Omega(x) := \{M \mid g(x, M) \leq p\}.$$

for some threshold $p$. This formulation enables us to capture input-dependent uncertainty sets in a principled and learnable way.

For PICNN [3], the score function $g$ is defined as

$$g(x, M) = W_L \sigma_L + V_L M + b_L,$$

where the internal layers are computed recursively as follows:

$$u_{l+1} = \text{ReLU}(R_l u_l + r_l), \quad u_0 = x,$$
$$\sigma_{l+1} = \text{ReLU}(W_l \sigma_l + V_l M + b_l), \quad \sigma_0 = 0,$$
$$W_l = \bar{W}_l \, \text{diag}([\hat{W}_l u_l + \omega_l]_+),$$
$$V_l = \bar{V}_l \, \text{diag}(\hat{V}_l u_l + v_l),$$
$$b_l = \bar{B}_l u_l + \bar{b}_l,$$

for all layers $l = 0, \ldots, L - 1$.

Therefore, the full set of parameters for the PICNN model is given by:

$$\theta^{\text{PICNN}} = \left\{ R_l, r_l, \bar{W}_l, \hat{W}_l, \omega_l, \bar{V}_l, \hat{V}_l, v_l, \bar{B}_l, \bar{b}_l \mid l = 0, \ldots, L \right\}.$$

Note that, in certain cases, the inner maximization problem of the original problems with PICNN-parametrized uncertainty sets may be infeasible or unbounded ones. This problem stems from too small a chosen $q$ (making $\Omega(x)$ empty) or $\Omega(x)$ is not compact. To address this concern, we modify PICNN architecture to ensure its sublevel sets are compact and introduce a slack variable to prevent it from being empty. Such modifications won't alter the general form of our problem.

## C  Proof of Proposition: Coverage with Quantile

We present a standard proof for the proposition 1 here:

**Proposition 1** (coverage with quantile) *Let the dataset $\mathcal{D} = \{(x_n, M_n)\}_{n=1}^N$ to be sampled i.i.d from the implicit distribution $\mathcal{P}$ gained during training phase. And $q$ is set to be the $(1-\alpha)$-quantile for the set $\{g(x_n, M_n)\}_{n=1}^N$, then $\Omega(x)$ gains the following guarantee.*

$$1 - \alpha \leq \mathbb{P}_{x,M,\mathcal{D}}(M \in \Omega(x)) \leq 1 - \alpha + \frac{1}{N+1}$$

*proof.* Let $\mathcal{X} = (x_i)_{i=1}^N$, $\mathcal{M} = (M_i)_{i=1}^N$, $g_i = g(x_i, M_i)$ for $i = 1, ..., N$, and $\mathcal{G} = g(\mathcal{X}, \mathcal{M})$. To avoid handling ties, assume the $g_i$ are distinct with probability 1. We begin by proving the lower bound of the inequality.

Without loss of generality, sort the calibration scores such that $g_1 \leq ... \leq g_N$. In this case, we have that $\hat{q} = g_{\lceil (n+1)(1-\alpha) \rceil}$ when $\alpha \geq \frac{1}{n+1}$ and $\hat{q} = \infty$ otherwise, where $\lceil \cdot \rceil$ denotes the ceiling operation. Assume $\alpha \geq \frac{1}{n+1}$. Observe the equivalence of the following two events:

$$\{\mathcal{M} \in \Omega(\mathcal{X})\} = \{\mathcal{G} \leq \hat{q}\}$$

Using the definition of $\hat{q}$, we have:

$$\{\mathcal{M} \in \Omega(\mathcal{X})\} = \{\mathcal{G} \leq g_{\lceil (n+1)(1-\alpha) \rceil}\}$$

Then, due to the exchangeability of variables in $(x_i, M_i)$ for $i = 1, ..., N$, the probability of $\mathcal{G}$ falling below a specific $g_k$ is:

$$P(\mathcal{G} \leq g_k) = \frac{k}{N+1}$$

In other words, $\mathcal{G}$ is equally likely to fall in anywhere between the calibration points $g_1, ..., g_N$. Note that above, the randomness is over all variables $g_1, ..., g_N$. From here, we next conclude:

$$P(\mathcal{G} \leq g_{\lceil (n+1)(1-\alpha) \rceil}) = \frac{\lceil (n+1)(1-\alpha) \rceil}{N+1} \geq 1 - \alpha$$

Thus, the lower bound is proven.

For the upper bound, we assume the conformal score distribution is continuous to avoid ties. The proof follows a similar process to the one of the lower bound. $\square$

## D  Derivation of the Dual Problems

We begin by defining the following notations: $\mathbf{0}_V$ denotes a zero vector of length $V$. $\mathbf{I}_{V \times V}$ and $\mathbf{0}_{V \times V}$ represents the $V \times V$ identity matrix and all-zero matrix, respectively.

The decision variable in the inner maximization problem is $M_i$, while $p_t$ is treated as a constant. We first rewrite the PICNN calibration constraint here for each $i$ and $v$.

$$g_i(x^i, M^i) \leq q, \ \forall i \in n \tag{8}$$

This can be equivalently reformulated as follows. Similarly, the indexes $i$ and $v$ are omitted for simplicity.

$$\begin{aligned}
&\sigma_l \geq \mathbf{0}_d, \ \forall l = 1, ..., L \\
&\sigma_{l+1} \geq W_l \sigma_l + V_l M + b_l, \ \forall l = 0, ..., L - 1 \\
&W_L \sigma_L + V_L y + b_L \leq q
\end{aligned} \tag{9}$$

To see why this holds, observe Eq. (9) is a relaxed form of Eq. (8), derived by replacing the equality constraints $\sigma_{l+1} = \text{ReLU}(W_l \sigma_l + V_l M + b_l)$ in the definition of the PICNN with two separate inequalities $\sigma_{l+1} \geq \mathbf{0}_d$ and $\sigma_{l+1} \geq W_l \sigma_l + V_l M + b_l$, for each $l = 0, ..., L-1$. Consequently, the optimal value of the relaxed problem cannot be less than that of the one of the original problem. The relaxed constraint Eq. (9) can be expressed in the following matrix form, where $A \in \mathbb{R}^{(2Ld+1) \times (V+Ld)}, b \in \mathbb{R}^{2Ld+1}$.

$$A \begin{bmatrix} M & \sigma_1 & \dots & \sigma_L \end{bmatrix}^T \leq b$$

$$A = \begin{bmatrix} & & -I_d & & \\ & & & \ddots & \\ & & & & -I_d \\ V_0 & -I_d & & & \\ \vdots & W_1 & \ddots & & \\ \vdots & & \ddots & -I_d & \\ V_L & & & W_L & \end{bmatrix}, \quad b = \begin{bmatrix} \mathbf{0}_d \\ \vdots \\ \mathbf{0}_d \\ -b_0 \\ \vdots \\ -b_{L-1} \\ q - b_L \end{bmatrix} \tag{10}$$

We first use the [U-MON] problem as an example and re-index $M$ with $i, v$ to ensure the completeness of the optimization problem. Additionally, we decompose the matrix $M_i$ into vectors $M_i^v$ for $v = 1, ..., V$. We denote the vector $[M_i^v \ (\sigma_1)_i^v \ ... \ (\sigma_L)_i^v]^T \in \mathbb{R}^{V+Ld}$ as $k_i^v$.

To this end, the uncertainty sets of the two-stage robust optimization problem are transformed into linear constraints. We reformulate the optimization problem as follows:

$$[\text{U-MON}] \min_{p_t} \max_{k_i^v} -\sum_{t \in T} \gamma^t \sum_{i \in n} (\beta_i^t)_0$$

$$s.t. \quad (1)$$
$$\alpha_i^t, \beta_i^t \in [0, 1], \forall i \in n, t \in T$$
$$T_v \alpha_i^t = (Hk_i^v)^T S \beta_i^{t-1}, \forall t \in T \setminus \{0\}, i, v \tag{11}$$
$$E\beta_i^t = 1, \forall i \in n, t \in T$$
$$\beta_i^0 = c, \forall i \in n$$
$$S\beta_i^t = D^t \alpha_i^t, \forall i \in n, t \in T$$
$$A_i^v k_i^v \leq b_i^v, \forall i \in n, v \ni V$$

where

$$S = \begin{bmatrix} \mathbf{0}_V & \mathbf{I}_{V \times V} \end{bmatrix} \in \mathbb{R}^{V \times (V+1)}$$

$$E = \begin{bmatrix} 1 & 1 & \dots & 1 \end{bmatrix} \in \mathbb{R}^{1 \times (V+1)}$$

$$H = \begin{bmatrix} \mathbf{I}_{V \times V} & \mathbf{0}_{V \times Ld} \end{bmatrix} \in \mathbb{R}^{V \times (V+Ld)}$$

$$D^t = \begin{bmatrix} d(p_t)_1 & & \\ & \ddots & \\ & & d(p_t)_V \end{bmatrix} \in \mathbb{R}^{V \times V}$$

$$(T_v)_i = \begin{cases} 1, & \text{if } i = v \\ 0, & \text{otherwise} \end{cases}, \quad T_v \in \mathbb{R}^{1 \times V}$$

are constant matrices for the inner maximization problem.

Assuming that the problem has optimal solutions and satisfies the Slater condition, we can invoke strong duality. Let $\pi_{i,v}^t$, $\mu_i$, $\lambda_i^t$, $\nu_i^t$, and $\xi_i^v$ represent the dual variables. The inner maximization problem can then be reformulated as an equivalent minimization problem using the KKT conditions and Lagrangian duality. Combining this reformulated minimization problem with the outer minimization, we derive the formulation in the main paper. where $C \in \mathbb{R}^{(V+Ld) \times (V+1)}$ and $\Gamma^t \in \mathbb{R}^{1 \times (V+1)}$ represent $H^T S$ and $\begin{bmatrix} \gamma^t & 0 & \dots & 0 \end{bmatrix}_{1 \times V} S$, respectively.

Similarly, the dual form [S-MON$^*$] of the [S-MON] problem can be derived as follows.

$$[\text{S-MON}^*] \min_{p,\pi,\mu,\lambda,\nu,\xi,k} \sum_{i,t,v} -c^T\mu_i - \lambda_i^t - (b_i^v)^T\xi_i^v + \Delta p_t$$

$$\begin{aligned}
s.t. \quad & (1) - (3) \\
& \Delta p_t = ||p_{t+1} - p_t||_2, \ \forall t \in T \setminus \{0\} \\
& D_t^T \nu_i^0 \le 0, \ \forall i \in n \\
& D_t^T \nu_i^t - \sum_{v \in V} \pi_{i,v}^t T_v^T \le 0, \forall i \in n, \ t \in T \setminus \{0\} \\
& \sum_{t \in T} \pi_{i,v}^t C \beta_i^t - (A_i^v)^T \xi_i^v \le 0, \xi_i^v \ge 0, \ \forall i, v \\
& (\Gamma^t)^T + (\nu_i^t)^T S + \lambda_i^t E + G_{i,t} \ge 0, \forall i \in n, \ t \in T \\
& G_{i,t} = \begin{cases} \mu_i^T - \sum_{v \in V} \pi_{i,v}^0 (k_i^v)^T C, & \text{if } t = 0 \\ 0, & \text{if } t = \tau \\ -\sum_{v \in V} \pi_{i,v}^t (k_i^v)^T C, & \text{otherwise} \end{cases}
\end{aligned}$$

# E   Convergence and Optimality of NEURO

**Convergence and Gradient Consistency**

Network components, parameterized by $\theta$, are trained using a composite reward signal derived from both optimization-specific rewards ($r_t^{\text{task}}$) and non-optimization rewards ($r_t^{\text{env}}$). Convergence to a desirable $\theta$ requires consistent gradients from these disparate sources.

We establish gradient consistency as follows. The gradient $\nabla_\theta r_t^{\text{task}}(\theta)$, originating from the convex Robust Optimization (RO) problem, inherently directs parameter updates towards configurations of $\theta$ that enhance the RO problem's objective value, thus steering $\theta$ towards an optimal parameter set $\theta^*$. Similarly, the gradient $\nabla_\theta r_t^{\text{env}}(\theta)$ from non-optimization objectives (e.g., generation quality) is engineered to guide $\theta$ towards the same $\theta^*$. The well-behaved nature of $\nabla_\theta r_t^{\text{env}}(\theta)$ is maintained through techniques including gradient clipping and normalization, ensuring its boundedness and local consistency.

Since both $r_t^{\text{task}}$ and $r_t^{\text{env}}$ are designed to drive $\theta$ towards a common (near) globally optimal $\theta^*$, their respective gradients, $\nabla_\theta r_t^{\text{task}}(\theta)$ and $\nabla_\theta r_t^{\text{task}}(\theta)$, are expected to exhibit consistent alignment. This directional alignment implies that updates suggested by each gradient component are collaborative rather than contradictory, which is critical for stable and effective learning using gradient-based methods. For applications involving non-convex or non-linear problem structures, NEURO's framework can be extended by incorporating PICNNs to maintain output convexity, or by integrating advanced gradient estimators and convex relaxations.

Beyond theoretical analysis, we empirically address gradient consistency in this multi-objective reinforcement learning setting by employing the Goal Vector Method (GVM) to combine $r_t^{\text{task}}$ and $r_t^{\text{env}}$. GVM formulates each reward component as a dimension in a goal vector, and adaptively projects the multi-dimensional gradient onto a unified descent direction. This projection balances task and environment objectives while ensuring that parameter updates lie within a consensus direction that respects both reward signals. By avoiding naive scalarization, GVM mitigates gradient interference between competing objectives and facilitates more stable convergence during training.

**Optimality Guarantees**

NEURO's convergence to the optimal solution of the *final, learned* RO problem is guaranteed under two readily satisfied assumptions: (i) The RO problem, formulated with the uncertainty set $\mathcal{U}(\theta)$ generated by the converged network parameters $\theta^*$, adheres to the structure of a Disciplined Convex Program (DCP). (ii) The objective and constraint functions within this RO problem are differentiable with respect to the decision variables and any parameters influenced by $\theta$.

Under these assumptions, and bolstered by the gradient consistency established above (which ensures the convergence of $\theta$ to a stable $\theta^*$), the RO problem defined by $\mathcal{U}(\theta^*)$ is convex. Consequently,

standard convex optimization solvers can efficiently identify its global optimum, $x^*(\theta^*)$. NEURO is thus guaranteed to converge to the optimal solution for this specific, data-driven RO formulation.

While the learning process for $\theta$ (which defines the uncertainty set $\mathcal{U}(\theta)$) navigates a potentially non-convex landscape—meaning $\mathcal{U}(\theta^*)$ itself may not be the globally "true" uncertainty set in an absolute sense—our framework guarantees optimality *for the RO problem constructed with $\mathcal{U}(\theta^*)$*. Crucially, due to our calibration guarantees, the learned uncertainty set $\mathcal{U}(\theta^*)$ effectively encapsulates a $(1 - \alpha)$ confidence region for the underlying uncertain parameters. Therefore, the solution $x^*(\theta^*)$ derived from NEURO represents the optimal worst-case performance over this empirically validated, high-confidence uncertainty region.

## F   Scalability of NEURO

In this section, we discuss the scalability of the NEURO framework, focusing on its runtime and task performance under larger grid configurations. We first examine the performance of the standard NEURO formulation and then introduce a method to enhance its computational efficiency for larger-scale problems.

**Baseline Scalability**

As shown in Table 3, while the absolute solving time for the optimization model remains manageable for moderately sized problems, it exhibits noticeable growth with $E$ and $\tau$ (this scalability trend and absolute solving time can be easily verified by constructing optimization problems of similar scale and solving them with Python library `CVXPY`). For typical indoor robot navigation, $E = 20$ might suffice. However, for applications demanding larger fields of view (e.g., autonomous vehicles), the optimization grid at this scale may not adequately represent real-world environments with fidelity. The computational complexity of the navigation task's optimization component is $\mathcal{O}(E^2\tau)$, indicating a quadratic dependence on the grid dimension $E$. Given that $E$ often has a more significant impact on navigation performance than $\tau$, in practice, one might fix $\tau$ to a smaller constant, resulting in a complexity of $\mathcal{O}(E^2)$. To effectively scale to larger environments, it is crucial to mitigate this quadratic growth associated with iterating over $E^2$ grid cells.

**Improving Scalability via Basis Function Expansion**

To address the scalability challenge, we explore the use of basis function expansion and sparse kernel approximation. Specifically, we represent the optimization variables $\alpha_i^t$ and $\beta_i^t$ (representing aspects of the utility or value functions over the grid) via a predefined or network-learned basis function decomposition $\{\phi_k(x, y)\}_{k=1}^K$:

$$\alpha_i^t = \sum_{k=1}^K a_k^t \phi_k(x, y), \quad \beta_i^t = \sum_{k=1}^K b_k^t \phi_k(x, y). \tag{12}$$

This representation subsequently modifies the original problem constraints. For instance, a constraint involving these variables, such as constraint (2b) from the main paper, can be reformulated based on the basis functions as follows:

$$a_k^t = \sum_{l=1}^K \left( \iint M^i(u, v)\phi_l(u, v)\phi_k(u, v)dudv \right) b_l^{t-1}. \tag{13}$$

The integral term in Eq. (13), $\iint M^i(u, v)\phi_l(u, v)\phi_k(u, v)dudv$, can be computed before solving with given $M^i$ and $\phi_k$. At this point, the optimization grid size is determined by the dimensionality of basis function vector $(a_k^t, b_k^t)$—predicting high-dimensional vectors is computationally inexpensive for neural networks. This adjustment reduces the time complexity of the underlying optimization module (referred to as `U-MON` in internal development) from $\mathcal{O}(E^2\tau)$ to $\mathcal{O}(K\tau)$, where $K \ll E^2$ is the number of basis functions. Empirically, this approach achieved a $23.6\times$ speedup (e.g., $0.023s$ inference time at $\tau = 8, E = 15$) in specific configurations, enabling consideration of significantly larger effective $E$.

Table 6 presents the performance of NEURO when employing this basis function expansion. The models were trained for 250k updates, consistent with the original experiments.

Table 6: NeuRO performance with basis function expansion. ($\texttt{S-MON}_{m=2}$)

| $E$ | Success(%) | Progress (%) | SPL(%) | Inference Time (s) |
|---|---|---|---|---|
| 20 | 82 | 85 | 76 | 0.251 |
| 50 | 83 | 85 | 77 | 0.670 |
| 100 | 85 | 87 | 80 | 1.310 |

The results in Table 6 demonstrate that the optimization problem's solving time now scales almost linearly with $E$ (effectively, with $K$ which might grow slowly with $E$, or $E$ itself if $K$ is chosen proportional to $E$ rather than $E^2$). However, we observe no substantial improvement in navigation performance for these larger $E$ values. We attribute this to two primary factors: (i) for typical indoor navigation tasks, the limited field of view and task characteristics may mean that smaller optimization grids ($E \approx 20$) are already sufficient to capture the necessary environmental information; (ii) the newly introduced module for predicting basis function coefficients ($a_k^t, b_k^t$) may require dedicated training strategies, more extensive data, or further architectural refinement to fully realize its potential.

In summary, this exploration offers a promising direction for enhancing the scalability of theNEUROframework. Nevertheless, we consider the primary contribution of this work to be the introduction and validation of the coreNEUROframework. This discussion on scalability is therefore presented as supplementary material, highlighting potential avenues for future research and application to more demanding, large-scale scenarios.

## G  Broader Application: Power Grid Scheduling as a Case Study

We recommend adopting the NEURO framework in the following scenarios. *First*, data-scarce environments where task performance is critical, as the optimization model can effectively capture general task rules that are intuitive to humans. *Second*, networked systems characterized by multi-stage, multi-objective, multi-agent, and multi-constraint structures, where optimization-based models offer a natural advantage in globally coordinating multiple entities. In this section, we study a representative application in the power market: the capacity expansion problem.

### G.1  Problem Setup.

The problem we address is a classic one in the context of power markets: capacity generation. The background of the problem is as follows: To enhance energy utilization efficiency, the power utility has deployed additional battery storage devices at each node in the grid. Our task, as the power utility, is to determine the optimal grid scheduling strategy to minimize overall electricity costs. A key challenge in this process is the uncertainty of electricity purchase prices, which introduces variability into the decision-making process.

However, we propose the following modifications to the original NEURO framework, see Fig. 5: *First*, since long-horizon planning is not required, we utilize a simpler deep learning structure instead of deep reinforcement learning. *Second*, as scheduling decisions are only meaningful under accurate power prices, we incorporate the loss between the network's predicted values and the ground-truth values into the overall loss function.

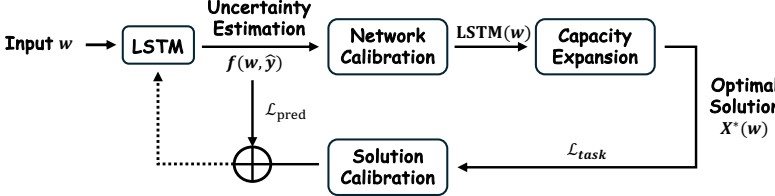

Figure 5: The workflow of solving capacity expansion problem using the NEURO framework.

**Network Component**

We generated a simple dataset $\mathcal{D} = \{(w_n, y_n)\}_{n=1}^M$, which contains weather data $w_n$ and the corresponding power prices $y_n$ over $M$ time points. The weather data $w_n = (h_n, s_n, p_n)$ include humidity $h_n$, sunlight intensity $s_n$, and precipitation intensity $p_n$. The network module consists of an LSTM network capable of handling non-uniform step-size inputs, followed by a PICNN. To simplify the problem, the predicted uncertainty of power prices is represented as upper and lower bounds (i.e., box uncertainty). During the $i$-th training step, the network takes historical electricity prices and corresponding weather data over the past $K$ time steps as input, $x_i = \{(w_j, y_j)\}_{j=i-K+1}^i$, and predicts electricity prices $\hat{y}_i = \{(\hat{y}_j^L, \hat{y}_j^H)\}_{j=1}^T$ for the next $T$ time steps. $\hat{y}_j^L$ and $\hat{y}_j^H$ represent the upper and lower bound, respectively. The prediction loss is defined as the mean squared error (MSE) loss between the uncertainty mean and the ground truth.

$$\mathcal{L}_{pred} = \sum_{j=1}^T \text{MSE}\left(\frac{\hat{y}_j^L + \hat{y}_j^H}{2}, y_{i+j}\right) \tag{14}$$

**Optimization Model**

Our power grid model adopts the 3-phase lindist flow model. In the lin-dist flow power grid model, the grid structure is assumed to be a tree. We consider $N$ nodes, where each node $i$ and its downstream nodes form a subtree $T_i$. Let $P_i$ represent the set of all paths from node $i$ to any downstream node. As the power utility, the decision variables include the complex power $s$ and voltage matrix $v_i$ at each node, with the voltage represented by a voltage matrix. The objective of the optimization is to minimize the total electricity procurement cost. Given the tree structure, this is equivalent to minimizing the product of the actual power supplied by the slack bus (node 0) and the electricity price. The optimization model can be formulated as follows.

$$\min_{s,v} \max_{\hat{y}_t \in [\hat{y}_t^L, \hat{y}_t^H]} \sum_{t=1}^T \hat{y}_t \text{Re}(s_0^t) \tag{15a}$$

$$s.t. \quad \sum_{j \in N} s_j^t = 0 \tag{15b}$$

$$\lambda_{ij}^t = -\sum_{k \in T_i} s_k^t, \ S_{ij}^t = \gamma \cdot \text{diag}(\lambda_{ij}^t) \tag{15c}$$

$$v_j^t = v_0^t - \sum_{(j,k) \in \mathcal{P}_i} (z_{jk} S_{jk}^{tH} + S_{jk}^t z_{jk}^H) \tag{15d}$$

$$s_j^{min} \le s_j^t \le s_j^{max} \tag{15e}$$

$$v_j^{min} \le \text{diag}(v_j^t) \le v_j^{max} \tag{15f}$$

$$\text{Re}(s_j^{b,t}) + \text{Re}(s_j^{d,t}) = \text{Re}(s_j^t) \tag{15g}$$

$$\text{SOC}_j^{t+1} = \text{SOC}_j^t + \text{Re}(s_j^t) \tag{15h}$$

$$\text{SOC}_j^0 = 0, \ \text{SOC}^{min} \le \text{SOC}_j^t \le \text{SOC}^{max} \tag{15i}$$

In the aforementioned model, $\text{Re}(\cdot)$ denotes the real part operator, $(\cdot)^H$ represents the Hermitian matrix operator, and $\text{diag}(\cdot)$ refers to the diagonalization operator. Eq. (15b) to Eq. (15d) define the power flow constraints in the lindist flow model, where $\gamma$ is a constant phase matrix, $z_{jk}$ represents the impedance matrix between nodes $j$ and $k$, and $S_{jk}^t$ and $\lambda_{jk}^t$ are intermediate variables. c15e and Eq. (15f) specify the operational constraints for each node. Eq. (15g) describes the composition of the power injection $s_i$ at each node, where $s_b$ is the dispatch power value of the battery, and $s_d$ represents the fixed constant power consumption at the node. Finally, Eq. (15h) and Eq. (15i) impose the state-of-charge (SoC) constraints for the battery devices.

Assuming the problem has an optimal solution $z^* = (s^*, v^*)$, the task loss $\mathcal{L}_{task}$ is defined as the optimal value of the objective function corresponding to this solution.

Table 7: Comparison on the capacity expansion task.

| # | Learning Framework | Prediction MSE↓ | Task Loss↓ |
|---|---|---|---|
| 1 | ETO | 275.44 | 810.52 |
| 2 | **NEURO** | **287.32** | **757.81** |

**Experiments**

We generated a set of synthetic weather data and corresponding electricity prices for training the network. In the downstream optimization model, the power grid structure is assumed to be a binary tree with seven nodes. The weights of Loss $\mathcal{L}_{pred}$ and Loss $\mathcal{L}_{task}$ in the NEURO framework are set to $0.8$ and $0.2$, respectively. As a baseline, we adopted the traditional "estimate-then-optimize" (ETO) task-based framework, where the network and the optimization model are decoupled. In the ETO method, the network is first trained to predict electricity prices, and during the testing phase, the network-predicted prices are fed into the optimization model to compute the scheduling strategy.

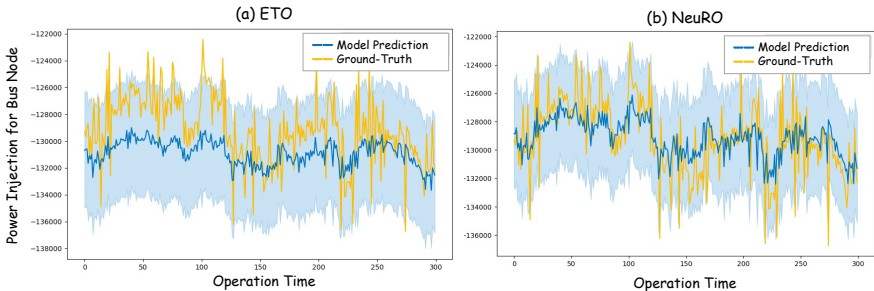

Figure 6: Visualization of prediction error on the capacity expansion problem.

The task loss and prediction loss for NEURO and ETO were recorded and are shown in Table 7, and the visualization of prediction loss can be found in Fig. 6 (the grey region is the visualization of the box uncertainty from the estimation). Table 7 indicates that NEURO's prediction accuracy is slightly lower than the traditional approach, but its task performance improves by 7%. Therefore, we believe NEURO's learning ability inherently involves a trade-off. However, in real-world applications, only accurate price predictions hold practical significance, making NEURO more suitable for tasks where precise recognition is not required. For example, in navigation, we care less about whether the robot forms human-like cognition and more about how it interacts with the environment to complete tasks.

## H   Impact of Target Object Geometry

We further investigate how the geometry of goal objects affects agent navigation performance in Table 8. We observe that agents find it easier to identify targets located near camera height (e.g., `Cylinder`) compared to taller objects that span the agent's full visual field. This suggests that object shape influences navigation success. Notably, NEURO significantly narrows this performance gap, likely due to its ability to encode general task-level information while being less sensitive to low-level visual cues such as material or shape. As a result, agents trained under the NEURO framework demonstrate improved generalization and adaptability across different task configurations.

Table 8: Navigation performance under different object geometries.

| Method | Cylinder | | Cube | | Sphere | |
|---|---|---|---|---|---|---|
| | Success | SPL | Success | SPL | Success | SPL |
| OracleEgoMap | 64 | 49 | 62 | 47 | 57 | 43 |
| Lyon | 76 | 62 | 74 | 57 | 73 | 57 |
| NEURO | **80** | **66** | **78** | **65** | **78** | **63** |

## I   Experimental Details

This appendix provides specific implementation details and parameter configurations for the key modules within the NEURO framework, complementing the descriptions in the main paper. The network components within the NEURO framework consist primarily of a Neural Perception Module and PICNNs.

For the Neural Perception Module, the architecture and parameters are adopted from the OracleE-goMap (Occ+Obj) configuration as described in the original MultiON work; for the PICNNs, we apply the following parameters:

Table 9: Hyperparameters for the Partially Input-Convex Neural Network (PICNN).

| Parameter | Value |
|---|---|
| Input dimension | 32 |
| Convex input dimension | 768 |
| Hidden layer dimension | 256 |
| Number of layers | 3 |
| Output dimension | $= V$ |
| Include $y$ in output layer | True |
| Feasibility parameter | 0.0 |

The 'output_dim' of $V$ is subsequently reshaped into a $E \times E$ grid, followed by a softmax operation along the last dimension, to produce the final output probabilities relevant to the navigation task.

All models were trained for 250,000 updates. Training was conducted on a single NVIDIA Quadro RTX 8000 GPU.

