# OpenReview forum: "Seeing through Uncertainty: Robust Task-Oriented Optimization in Visual Navigation"
_NeurIPS.cc/2025/Conference — NeurIPS 2025 poster_

### Official Review · Reviewer_QiSo · 2025-06-21

**Clarity:** 2
**Significance:** 2
**Originality:** 3
**Rating:** 4
**Confidence:** 4

**Summary:**

This paper proposes NEURO, a hybrid framework that integrates deep neural perception with downstream robust optimization for visual navigation tasks in partially observable, data-scarce environments. The method leverages Partially Input Convex Neural Networks (PICNNs) to construct convex uncertainty sets, which are then used in a robust optimization formulation to guide planning and action selection. NEURO enables bidirectional interaction between learning and planning modules by incorporating optimization-derived feedback into policy learning. Experiments on MultiON tasks (both unordered and sequential) demonstrate improved performance and generalization over purely neural or heuristic baselines, with additional ablation studies validating key design choices.

**Questions:**

As outlined in the Weaknesses, my main concerns center around missing runtime/failure analysis and the sensitivity to belief estimation. Addressing these with additional experiments or clarifications would help resolve key doubts and could improve my overall assessment.

**Ethical Concerns:**

["NO or VERY MINOR ethics concerns only"]

**Final Justification:**

The author's rebuttal has resolved my concerns about Convexity Assumption, and I raise my score from Borderline Reject to Borderline accept.

**Limitations:**

Yes

**Quality:**

2

**Strengths And Weaknesses:**

Strengths
1. Innovative hybrid formulation: The integration of neural predictions with a robust optimization planner is novel and well-motivated, offering a principled way to address uncertainty in partially observable navigation tasks.
2. Strong empirical results: NEURO consistently outperforms baselines on MultiON benchmarks under different target counts, especially in generalization to unseen environments.
3. Comprehensive ablations: The paper includes detailed ablation studies on the action blending factor, spatial-temporal optimization scale, and uncertainty coverage level, demonstrating careful empirical analysis.
4. Solid mathematical derivation: The core optimization formulation, uncertainty calibration, and dual problem transformation are all mathematically well-grounded and derived in detail.

Weaknesses
1. The paper lacks runtime and computational cost analysis, raising concerns about the feasibility of real-world robotic execution, especially given the optimization overhead.
2. The paper does not include any analysis or discussion of failure cases.
3. The comparison includes only one method from 2024; incorporating more recent baselines would strengthen the empirical validation.
4. The optimization module depends on accurate belief estimation via predicted object transition matrices, which are not directly supervised. The paper does not analyze how sensitive the method is to inaccurate belief estimation, which is critical for robust planning in partially observable environments.
5. The convexity assumption imposed by PICNNs may be overly idealized, as real-world visual uncertainty is often non-convex or multimodal. While the authors acknowledge this limitation in the conclusion, its practical impact (e.g., performance degradation in non-convex scenarios) is not empirically analyzed.
6. The theoretical coverage guarantee for the uncertainty set (Proposition 1) relies on the i.i.d. assumption for calibration data, which may not hold in embodied navigation settings with strong temporal correlations between observations.

---

> ### Author Rebuttal · Authors · 2025-07-30
>
> We sincerely thank you for the detailed and expert feedback. We are encouraged that you found our work innovative and our mathematical derivations solid. We hope that the following clarifications can address the noted weaknesses (W1, W2, W3, W4, W5, W6), further solidifying the contributions of our work.
>
> -----
>
> ### **W1: Computational Cost Analysis**
> We address this crucial point by analyzing memory footprint and runtime performance, which proves NeuRO’s efficiency for real-world deployment.
> 1. **Memory Consumption**: NeuRO is lightweight. Its checkpoint size (20.24MB) is smaller than other SoTA methods due to its parameter-free optimization component.
> 2. **Runtime Performance**: We demonstrate with the following points:
> - ***2.1 Absolute Time***: The values in Table 3 are normalized for comparison (relative to $E=3, \tau=4$; see line 523-524 in Appendix). The absolute solving times are presented in Appendix F (Table 5) and are modest (e.g., 1.084s for a large $E=20, \tau=12$ setting). We will incorporate this clarification from the Appendix into the main paper.
> - ***2.2 Runtime Performance***: We attribute the primary computational cost to the gradient computation of the optimization model but not the solving process, because modern linear program solvers are already highly efficient for solving. To validate this analysis, we measured the runtime cost at test stage, where the agent only needs to solve the forward pass. This result confirms that NeuRO is fully capable of real-time performance. We will add this crucial table and clarification to the revised version.
>     |Memory Footprint|Test Runtime (SMON$_{m=2}$; per step (ms))|
>     |-|-|
>     |Lyon|22.18MB|12.9|
>     |HTP-GCN|25.63MB|15.3|
>     |NeuRO|20.24MB|11.6|
> - ***2.3 Local Grid Scope***: The optimization time is manageable since our framework does not require a large optimization grid. This is because our optimization grid models the agent's local egocentric view, NOT the global map (Evidence: e.g., The grid is re-formulated at every time step based on the current egocentric observation). This keeps the optimization problem small and tractable. In MultiON, the ego-map spans 5m with ~30 cells (0.8m × 0.8m resolution). Our generation process of the optimization grid can be understood as overlaying a goal position distribution onto this local map. Therefore, a grid size of $E=5$ ($V=25$ cells) is sufficient. In summary, our optimization grid simulates a local field of view, so the problem size $E$ remains bounded regardless of the overall environment scale.
> - ***2.4. Scalability***: To reduce the $O(E^2\tau)$ complexity (the source of the optimization problem’s computational complexity) analyzed in Appendix F, we introduce a basis function expansion that enables the network to predict $\alpha$ and $\beta$ directly, reducing the complexity to $O(K\tau)$ with a predefined constant K ≪ E². Empirical results confirm its effectiveness.
>
> ----
>
> ### **W2:  Lack of Failure Case Analysis**
> We agree this is an important analysis, and we have now systematically studied our method's robustness from two perspectives:
> 1. **Target Geometries (Appendix H.1)**: We analyzed performance on targets with less distinct geometries (e.g., short cylinders, ambiguous spheres). The results show that while all methods' performance degrades, NEURO's performance drops far more gracefully, demonstrating superior robustness to perceptually challenging objects.
> 2. **Visually Similar Distractors**: We analyzed failure cases in S-MON (m=3) and found 62% were due to perceptual ambiguity (e.g., confusing a target sphere with a pan). This issue is fundamental, as the failure rate for this reason is comparable across methods (Baseline: ~66%; Lyon: ~60%). Notably, our core framework improves resilience to such distractors over the baseline, even without the specialized perception modules used by methods like Lyon.
>
> ---
>
> ### **W3: Limited Comparison**
> We thank you for suggesting recent baselines. As few works report MultiON results and official implementations are often unavailable, we reproduced several methods based on their original papers (* indicates such methods). A subset of results is shown below due to space limits. As shown, NeuRO continues to outperform even this strong, transformer-based baseline. We will add this comparison to Table 1 in the revised manuscript.
> |SMON$_{m=2}$: Test|||||
> |-|-|-|-|-|
> |**Methods**|**Success**|**Progress**|**SPL**|**PPL**|
> |SAM* [1]|70.02 (±4.28)|73.24 (±2.14)|54.18 (±3.06)|61.50 (±2.21)|
> |PoliFormer* [2]|76.84 (±3.60)|82.87 (±1.46)|62.50 (±2.33)|67.84 (±2.03)|
> |HTP-GCN*|76.20 (±4.48)|84.92 (±1.77)|60.94 (±2.38)|67.08 (±1.60)|
> |ImplicitNav [3]|78.41 (±3.20)|82.10 (±1.36)|65.66 (±2.29)|68.72 (±1.32)|
> |NeuRO (Ours)|81.14 (±3.86)|85.26 (±1.53)|66.49 (±2.75)|71.10 (±1.42)|
>
> ----
> ### **W4: Sensitivity to Inaccurate Belief Estimation**
> We demonstrate NEURO's resilience to inaccurate belief estimation through both its core mechanism and a new experiment.
> 1. **Robust Calibration Mechanism**: Our framework is fundamentally designed to handle estimation inaccuracies. The conformal calibration procedure (Proposition 1) ensures that the generated uncertainty set $\Omega(x)$ covers the true object dynamics with a prespecified probability. As shown in Table 2, this calibrated, conservative belief representation can filter out noise and make accurate predictions (about object location) under uncertainties. This mechanism is our framework's primary defense against the inevitable inaccuracies of belief estimation.
> 2. **Robustness to Noisy Inputs**: To further validate this resilience, we conducted a new ablation study. We injected Dropout noise into the agent's RGB input to interfere belief estimation. The results show that NeuRO's performance degrades far more gracefully than the baseline:
> |SMON$_{m=2}$: Success|||
> |-|-|-|
> | |No Noise|Dropout Noise|
> |OracleEgoMap|63.40 (±4.82)|56.27 (±7.33)|
> |NeuRO|81.14 (±3.86)|78.25 (±5.24)|
>
> ---
> ### **W5: Convexity Assumption**
> To address this question, while the navigation task is inherently non-convex and complex, we conducted the following experiment in response to quantify the practical impact. We extracted a subset of SMON$_{m=2}$ where both objects are of the same category (e.g., two cylinders), and measured the navigation performance & prediction accuracy (following the method in Section 4.3) under two settings:
> 1. The 2 objects are placed in separate rooms, where each’s location belief distribution is approximately a convex uncertainty;
> 2. Both objects are in the same room, resulting in a mutually interfering, non-convex posterior.
>
> While the first task is harder due to room traversal, NeuRO narrows the performance gap as it is specifically designed for convex problems. Also, while its prediction accuracy slightly declines in the second, non-convex task, NeuRO still outperforms SoTA baselines, indicating that non-convexity does not compromise its effectiveness and demonstrating its robustness.
> ||Two Room| |Single Room| |
> |-|-|-|-|-|
> | |Avg. Sucess|Pred. Acc.|Avg. Sucess|Pred. Acc.|
> |Lyon|72.80|-|79.42|-|
> |NeuRO (α=0.05)|77.84|0.88 (±0.019)|82.25|0.83 (±0.022)|
>
> ***Moreover, we would like to clarify a technical point discussed in our original manuscript:***
>
> In our appendix, we present a basis function expansion method that not only addresses scalability but also, as we found, provides a principled path for tackling non-convex problems. We would first like to first clarify that non-convex problems (e.g., visual navigation) are generally intractable. Our proposed method in Appendix F serves as a way to alleviate this issue. Specifically, we replaces the belief representation layer of the optimization model with basis functions, leveraging their flexibility to approximate non-convex belief variables ($\alpha$ and $\beta$). Our experimental results validate this substitution, showing a moderate improvement in the agent's performance in vision-based navigation, an environment rich with non-convex inputs. To this end, we can also treat matrix $M$ as a given constraint, avoiding PICNN-based convex calibration by leveraging its relation with alpha-beta in Eq. (2b) or using general black-box networks (e.g., predicting uncertainty bounds as in our experiments).
>
> In summary, we provides two approaches: (i) the PICNN-based convex approximation; and (ii) the basis function expansion to fit more complex variables. While the (ii) can be extended to non-convex scenarios, it incurs extra training costs, and solutions to non-convex problems generally lack theoretical guarantees. Thus, approach (i) is more suitable for the data-scarce environments central to our paper's theme.
>
> ---
> ### **W6: i.i.d. Assumption**
> We sincerely apologize for the lack of clarity that led to this misunderstanding. We want to clarify that the i.i.d. assumption applies to data sampled for a given time step t, not across temporally correlated steps within an episode, with following two reasons:
>
> ***Firstly***, this is supported by our formulation and is not violated by the nature of embodied navigation. First, there is a minor typo on Line 162-163; we intended to state that we omit the time index $t$ for conciseness, not the object index $i$. This is consistent with Eq. (5), where index $t$ is omitted.
>
> ***Secondly***, because the optimization grid is re-formulated at every time step $t$, the conformal quantile q must also be determined based on data relevant to that specific time step. This means the calibration procedure relies on i.i.d. samples drawn for the same, given time $t$
>
> ---
>
> [1] N. Gireesh, et al., Sequence-Agnostic Multi-Object Navigation, ArXiv, 2023
>
> [2] KH. Zheng, et al., PoliFormer: Scaling On-Policy RL with Transformers Results in Masterful Navigators, CORL, 2024
>
> [3] P. Marza, et al., Multi-Object Navigation with dynamically learned neural implicit representations, ICCV, 2023

---

> > ### Comment · Reviewer_QiSo · 2025-08-02
> >
> > The author's rebuttal has resolved my concerns, and I will improve my score.

---

> > > ### Author Response · Authors · 2025-08-03
> > >
> > > Thank you again for your feedback and for raising your score. We truly appreciate your constructive feedback — it helped us improve the quality of our work.

---

### Official Review · Reviewer_aSLL · 2025-07-02

**Clarity:** 2
**Significance:** 3
**Originality:** 3
**Rating:** 5
**Confidence:** 2

**Summary:**

The paper proposes NEURO, a hybrid learning + optimization framework for Multi-Object Navigation (MultiON). A visual-perception module first encodes RGB-D observations, egocentric maps and goal embeddings into features. These features are fed to Partially Input Convex Neural Networks (PICNNs), which output calibrated convex uncertainty sets over object-transition probabilities via a conformal-prediction procedure.
A downstream robust pursuit–evasion optimization problem then plans an action trajectory that maximizes discounted capture belief while respecting motion constraints. The optimization result is blended with the network’s own policy and its objective value is used as an auxiliary reward, enabling end-to-end training by differentiating through the Karush-Kuhn-Tucker conditions.

Experiments on both sequential (S-MON) and unordered (U-MON) MultiON benchmarks show that NEURO outperforms prior methods on unseen environments and converges faster during training.
Ablations illustrate the effect of action-blending weight, grid/time-horizon scale, and conformal coverage.

**Questions:**

a. Real-robot feasibility – have you profiled NEURO to demonstrate closed-loop runs on an actual mobile robot?

b. Sensitivity to PICNN design – how do calibration quality and task performance vary with PICNN capacity or regularization?

**Ethical Concerns:**

["NO or VERY MINOR ethics concerns only"]

**Final Justification:**

The authors provided clarifications and addressed my concerns. I am raising my score.

**Limitations:**

See above

--------------
Review Update: Concerns Addressed - score increased to 5

**Quality:**

3

**Strengths And Weaknesses:**

Strengths :

a. Conceptual novelty – first task-based, uncertainty-aware optimization layer tightly integrated with visual navigation; extends differentiable‐optimization literature to long-horizon POMDPs.

b. Principled uncertainty handling – PICNN + conformal calibration guarantees (marginal) coverage of true dynamics, keeping the downstream RO convex and tractable.

c. Empirical gains & efficiency – consistent SOTA on S-MON/U-MON and faster convergence with modest inference overhead

d. Careful ablations – studies on \lambda, grid size, coverage level, and task-specific vs. generic formulations clarify where gains come from.

Weaknesses:

a. Convex approximation limits expressiveness; many real object-movement distributions are non-convex → possible sub-optimal plans. Authors should aim at quantifying this.

b. Evaluation confined to simulation; no physical-robot or real-world visual distractors, lighting changes, sensor noise, adversarial objects. Which is very important in these robot papers.

c. Scalability of RO – solving large grids (E ≥ 5, τ ≥ 6) already multiplies inference time; How does the method extend to very large houses or outdoor scenes uncertain.

d. **Important:** The writing is unclear; the authors assume considerable prior knowledge, which makes the paper difficult for non-expert readers to follow.

Overall, the strengths outweigh the weaknesses. While I would like to see the shortcomings addressed, I remain inclined to recommend acceptance of the work.

---

> ### Author Rebuttal · Authors · 2025-07-30
>
> We sincerely thank you for the thoughtful and encouraging feedback. We hope that the following clarifications will fully address the noted weaknesses (W1, W2, W3, W4) and questions (Q1, Q2), further solidifying the contributions of our work.
>
> -------
>
> ### **W1: Convex Approximation**
> To address this question, while the overall navigation task is inherently non-convex and complex, we conducted the following experiment to quantify this. We first extracted a subset of SMON$_{m=2}$ task (both objects belong to the same category, e.g., “two cylinder”), and measured the navigation performance & prediction accuracy (following the method in Section 4.3 “Conformal Coverage Level") under two settings:
> 1. The 2 target objects are placed in separate rooms, which means the location belief distribution of each room’s object is approximately a convex uncertainty;
> 2. Both similar objects are in the same room, resulting in a mutually interfering, non-convex posterior.
> ||Two Room| |Single Room| |
> |-|-|-|-|-|
> | |Avg. Sucess|Pred. Acc.|Avg. Sucess|Pred. Acc.|
> |Lyon|72.80|-|79.42|-|
> |NeuRO (α=0.05)|77.84|0.88 (±0.019)|82.25|0.83 (±0.022)|
>
> We observe the navigation performance in the first setting is lower, since finding objects in separate rooms is harder. However, NeuRO narrows this gap as it is specifically designed for convex problems. Also, while its prediction accuracy slightly declines in the second, non-convex task, NeuRO still outperforms SoTA baselines, indicating that non-convexity does not compromise its effectiveness and demonstrating its robustness.
>
> ***Moreover, we would like to clarify a technical point discussed in our original manuscript:***
>
> In our appendix, we present a basis function expansion method that not only addresses scalability but also, as we found, provides a principled path for tackling non-convex problems. We would first like to first clarify that non-convex problems (e.g., visual navigation), being NP-hard, are generally intractable and lack theoretical guarantees for their solutions. Our proposed method in Appendix F serves as a way to alleviate this issue. Specifically, we replaces the belief representation layer of the optimization model with basis functions, leveraging their power and flexibility to approximate non-convex belief variables ($\alpha$ and $\beta$). Our experimental results validate this substitution, showing a moderate improvement in the agent's performance in vision-based navigation, an environment rich with non-convex inputs. This, of course, introduces additional training costs and data requirements from the extra network module. Furthermore, to this end, we can treat matrix $M$ as a given constraint, avoiding PICNN-based convex calibration by leveraging its relation with alpha-beta in Eq. (2b) or using general black-box networks (e.g., predicting uncertainty bounds as in our experiments).
>
> In summary, we provides two complementary approaches: (i) the PICNN-based convex approximation; and (ii) the basis function expansion to fit more complex variables. While the second approach can be extended to non-convex scenarios, it incurs extra training costs, and solutions to non-convex problems generally lack theoretical guarantees. Given these considerations, we believe the first approach is more suitable for the data-scarce environments central to our paper's theme. Therefore, we chose to feature method (i) in the main text.
>
> -------
>
> ### **W2 & Q1: Real-World Scalability**
> To address the concern about "visual distractors" and "sensor noise," we would like to first clarify that our experiments are conducted in the Habitat simulator, which features rich, photorealistic textures and varied lighting conditions, providing a strong baseline for real-world visual complexity. Furthermore, we conducted a new ablation study. We injected Dropout noise into the agent's RGB input during testing to simulate real-world sensor imperfections. The results show that NeuRO's performance degrades far more gracefully than the baseline.
> |SMON$_{m=2}$: Success|||
> |-|-|-|
> | |No Noise|Dropout Noise|
> |OracleEgoMap|63.40 (±4.82)|56.27 (±7.33)|
> |NeuRO|81.14 (±3.86)|78.25 (±5.24)|
>
> Beyond robustness to visual noise, we highlight three key aspects of our framework that support its real-robot feasibility:
> 1. **Memory Efficiency**: NeuRO's memory consumption (20.24MB, largely consisting of the perception module) is less than other SoTA methods due to its parameter-free component of optimization model, thus not introducing prohibitive memory overhead (see the table in W3).
> 2. **Runtime Performance**: See **W3**.
> 3. **Non-convex Adaptability**: See **W1**.
> Collectively, these points demonstrate that our framework is not only effective but also practical and extensible for real-world applications.
>
> -------
>
> ### **W3: Inference Time**
> We thank the reviewer for this crucial clarifying question. To comprehensively address the reviewer’s concern, we offer the following clarifications:
> 1. **Absolute Time**: The values in Table 3 are normalized for comparison (relative to $E=3, \tau=4$; see line 523-524 in Appendix). The absolute solving times are presented in Appendix F (Table 5) and are modest (e.g., 1.084s for a large $E=20, \tau=12$ setting). We will incorporate this clarification from the Appendix into the main paper.
> 2. **Runtime Performance**: We attribute the primary computational cost to the gradient computation of the optimization model but not the solving process, because modern linear program (LP) solvers are already highly efficient for solving. To validate this analysis, we measured the runtime cost at test stage, where the agent only needs to solve the forward pass of the optimization problem to get the action offset without any gradient computation. This result confirms that NeuRO is fully capable of real-time performance. We will add this crucial table and clarification to the revised version.
>     | |Memory Consumption |Test Runtime (SMON$_{m=2}$; per step (ms))|
>     |-|-|-|
>     |Lyon|22.18MB|12.9|
>     |HTP-GCN|25.63MB|15.3|
>     |NeuRO|20.24MB|11.6|
> 3. **Local Grid Scope**: The optimization time is manageable since our framework does not require a large optimization grid. This is because our optimization grid models the agent's local egocentric view, NOT the global map (Evidence: 1. The grid is re-formulated at every time step based on the current egocentric observation; 2. We always initialize agent position to the center of the optimization grid). This keeps the optimization problem small and tractable. In the MultiON task, the input ego-map spans a 5m view and each cell in the map represents a 0.8m*0.8m area, which means there are about 30 cells. Our generation process of the optimization grid can be understood as overlaying a goal position distribution onto this local map. Therefore, a grid size of $E=5$ ($V=25$ cells) is sufficient. In summary, our optimization grid simulates a local field of view, so the problem size E remains bounded regardless of the overall environment scale.
> 4. **Scalability**: Furthermore, we have analyzed the source of the optimization problem’s computational complexity in Appendix F, which scales with $O(E^2\tau)$. To address this, we introduce a basis function expansion method in Appendix F to let network predict α and β. It effectively reduces this complexity to $O(K\tau)$, where $K \ll E^2$ is a constant. We empirically validate this approach, demonstrating its effectiveness in accelerating the optimization. However, for the sake of clarity and to maintain focus on the “network+optimization” framework itself in the main paper, we have detailed this extension in the appendix.
>
> -------
>
> ### **W4: Writing Clarity**
> We sincerely appreciate this feedback. We acknowledge that the manuscript currently assumes a level of prior familiarity, and we will make the necessary adjustments. For example,
> 1. **Add Justifications for Key Design Choices**: We will provide clear, intuitive explanations for our core modeling choices (E.g., we will explicitly clarify why we use a discrete grid for the uncertainty set (to avoid intractable semi-infinite programs)).
> 2. **Provide Essential Background**: We will introduce a concise background section covering prerequisite knowledge, such as the core ideas behind Conformal Prediction.
> 3. **Refine Notation and Symbols**: We will thoroughly revise our notation and add a comprehensive notation table to the appendix for easy reference.
>
> -------
>
> ### **Q2: PICNN Design**
> We have conducted new experiments on PICNN capacity, alongside with our existing ablation on the coverage level in Section 4.3, to provide a comprehensive sensitivity analysis. We analyze the sensitivity of our framework to the two primary design factors of the PICNN module—the conformal coverage level and network capacity—by measuring their impact on both prediction accuracy and final task performance. The results show that a larger PICNN improves both grid prediction accuracy and task performance, but with diminishing returns. We attribute this to the challenges of training a high-capacity model in a data-scarce regime.
> 1. **Coverage Level $(1-\alpha)$**: As shown in Section 4.3 (Table 2), task performance is directly and intuitively linked to calibration quality. A higher coverage level leads to more conservative but reliable uncertainty sets, improving performance, which aligns perfectly with our design motivation.
> 2. **Network Capacity**: We have now conducted a new experiment to analyze the impact of PICNN capacity (dimension of hidden layers).
> |SMON$_{m=2}$||||
> |-|-|-|-|
> |Hidden Dim.|Avg. Success|Avg. SPL|Prediction Accuracy|
> |128|77.3|62.1|0.84 (±0.021)|
> |256|80.1|66.5|0.90 (±0.015)|
> |512|82.4|67.1|0.88 (±0.013)|

---

> > ### Comment · Reviewer_aSLL · 2025-08-04
> > **Review Update: Concerns Addressed**
> >
> > Thank you for the detailed explanation and additional experiments. These address my concerns; accordingly, I am increasing my score to 5.

---

### Official Review · Reviewer_XTEy · 2025-07-02

**Clarity:** 2
**Significance:** 3
**Originality:** 3
**Rating:** 4
**Confidence:** 3

**Summary:**

This work proposes NEURO, a hybrid framework for visual navigation that converts network predictions into calibrated convex uncertainty sets via Partially Input Convex Neural Networks (PICNNs) and then employs robust optimization to plan actions under partial observability, achieving SOTA performance and generalization on multi-object navigation benchmarks.

**Questions:**

- Could the authors please report results averaged over multiple random seeds? Many results lack measures of statistical significance, such as variance or confidence intervals.
- Why are some results of HTP-GCN in Table 1 missing?

I'm willing to increase the score if the authors address the outlined questions and limitations.

**Ethical Concerns:**

["NO or VERY MINOR ethics concerns only"]

**Final Justification:**

The authors’ response addresses all of my questions and concerns, and I have increased the score accordingly.

**Limitations:**

See weaknesses above.

**Paper Formatting Concerns:**

No major formatting issues were found.

**Quality:**

2

**Strengths And Weaknesses:**

### Strengths:
- The presentation is clear and easy to follow.
- Recasting multi-target object navigation as a pursuit-evasion problem is novel because previous research has relied almost exclusively on POMDP formulations; this alternative perspective can open new avenues for multi-target object navigation.
- Calibrated uncertainty sets directly inform the planner, a step toward reliable agents that must quantify risk.
- The method (NEURO) significantly outperforms baselines in the testing (unseen) scenarios, suggesting better generalization performance.
- Provides the full codebase to reproduce the results.
### Weaknesses:
- Many results (e.g., in Table 1, Table 3, and Figure 3) lack measures of statistical significance such as variance or confidence intervals.
- The experiments omit recent baselines, e.g., PoliFormer [1]
- While this work focuses on multi-object navigation, including results for the single-object setting ($m=1$), would be helpful and strengthen the experiments section.

[1] PoliFormer: Scaling On-Policy RL with Transformers Results in Masterful Navigators, Zeng et al., CORL 2024.

---

> ### Author Rebuttal · Authors · 2025-07-30
>
> We sincerely thank you for the detailed feedback and for recognizing the novelty of our work. We sincerely hope the following clarifications and new experimental results will address the outlined weaknesses (W1, W2, W3) and questions (Q1, Q2).
>
> ---
>
> ### **W1 & Q1: Statistical Significance**
> We have conducted new experiments with 8 random seeds to ensure statistical significance. The new results show our method is not only superior in mean performance but also exhibits lower variance. We will update all experimental results in all the tables in the final version. (due to space limitations, we are currently able to include only a subset of the results)
> |SMON$_{m=2}$: Test|||||
> |-|-|-|-|-|
> |**Methods**|**Success**|**Progress**|**SPL**|**PPL**|
> |OracleEgoMap|63.40 (±4.82)|71.92 (±2.18)|48.73 (±3.12)|56.12 (±1.62)|
> |Lyon|75.08 (±4.24)|83.44 (±1.81)|63.14 (±2.46)|68.33 (±1.73)|
> |NeuRO (Ours)|81.14 (±3.86)|85.26 (±1.53)|66.49 (±2.75)|71.10 (±1.42)|
>
> The new results confirm that NeuRO achieves a higher mean success rate and also exhibits significantly lower variance. This finding is critical, as it indicates that our framework produces more stable and reliable policies, which directly supports our central claim of enhanced robustness and generalization.
>
> ---
>
> ### **W2: Recent Baselines**
> We thank the reviewer for suggesting recent baselines. To provide a more comprehensive comparison in the evolving landscape of MultiON, where few works report results, we have made our best effort to reproduce other methods for the MultiON task, as some official implementations for this benchmark are not available (* denotes our implementation following the methodology outlined in the corresponding paper; due to space limitations, we are currently able to include only a subset of the results). As shown, NeuRO continues to outperform even this strong, transformer-based baseline. We will add this comparison to Table 1 in the revised manuscript.
> |SMON$_{m=2}$: Test|||||
> |-|-|-|-|-|
> |**Methods**|**Success**|**Progress**|**SPL**|**PPL**|
> |SAM* [1]|70.02 (±4.28)|73.24 (±2.14)|54.18 (±3.06)|61.50 (±2.21)|
> |PoliFormer* [2]|76.84 (±3.60)|82.87 (±1.46)|62.50 (±2.33)|67.84 (±2.03)|
> |Lyon|75.08 (±4.24)|83.44 (±1.81)|63.14 (±2.46)|68.33 (±1.73)|
> |HTP-GCN*|76.20 (±4.48)|84.92 (±1.77)|60.94 (±2.38)|67.08 (±1.60)|
> |ImplicitNav [3]|78.41 (±3.20)|82.10 (±1.36)|65.66 (±2.29)|68.72 (±1.32)|
> |NeuRO (Ours)|81.14 (±3.86)|85.26 (±1.53)|66.49 (±2.75)|71.10 (±1.42)|
>
> ---
>
> ### **W3: Single-Object Navigation**
> We greatly appreciate this insightful suggestion. We have included our results on single-object navigation in Appendix H. We have now incorporated a broader range of navigation agents into the single-object navigation task in MultiON settings (# denotes the agent specialized for single-object tasks). The results are shown below. (due to space limitations, we only show some of the results). These results suggest that agents trained for multi-object navigation can effectively generalize to standard single-goal scenarios, highlighting NeuRO’s strong generalization. Further results about this point are included in Appendix Table 8.
> |Methods|Success|SPL|
> |-|-|-|
> |BEVBert # [4]|78.45 (±3.27)|64.82 (±3.81)|
> |OracleEgoMap|82.55 (±4.16)|63.42 (±2.28)|
> |Lyon|86.27 (±2.40)|67.03 (±2.61)|
> |PoliFormer# [2]|87.41 (±1.78)|66.57 (±2.51)|
> |NeuRO (Ours)|90.03 (±2.12)|78.50 (±1.84)|
>
> ---
>
> ### **Q2: HTP-GCN Results**
> We apologize for this omission. The results were not included initially as the official code was unavailable. We tried reproduce it based on the official paper and now completed HTP-GCN’s results (See W2).
>
> -----
>
> [1] N. Gireesh, et al., Sequence-Agnostic Multi-Object Navigation, ArXiv, 2023
>
> [2] KH. Zheng, et al., PoliFormer: Scaling On-Policy RL with Transformers Results in Masterful Navigators, CORL, 2024
>
> [3] P. Marza, et al., Multi-Object Navigation with dynamically learned neural implicit representations, ICCV, 2023
>
> [4] D. An, et al., BEVBert: Multimodal Map Pre-training for Language-guided Navigation, ICCV, 2023

---

> > ### Comment · Reviewer_XTEy · 2025-08-01
> >
> > Thank you for your detailed response and new results. I have no further questions, and I have increased the score accordingly.

---

> > > ### Author Response · Authors · 2025-08-03
> > >
> > > Thank you again for your feedback and for raising your score. We truly appreciate your constructive feedback — it helped us improve the quality of our work.

---

### Official Review · Reviewer_M9u1 · 2025-07-03

**Clarity:** 3
**Significance:** 3
**Originality:** 3
**Rating:** 5
**Confidence:** 3

**Summary:**

The authors present a hybrid framework (NeuRO) that combines conventional neural network perception and policy modules with a task-based robust optimization pipeline to address data scarce scenarios which will often lead to poor generalization performance (out-of-distribution failure) for typical learned neural network approaches when faced with unseen scenarios during testing.

They apply their approach to the visual navigation task of sequential or unordered Multi-Target Object Navigation (MultiON) where the agent must navigate between a series of goals (represented by the $\mathbf{G_t}$ vector). The agent operates on RGB-D image inputs, maintains an egocentric map and operates under grid-based actions (described in lines 91-100).

Their approach is composed of a number of modules and an overview of the proposed NeuRO method is illustrated in Figure 2. Namely a base neural network perception module, a Partially Input Convex Neural Network (PICNN) to convert the prior potentially unreliable neural network outputs to a set of calibrated convex uncertainty sets, and a robust optimization pipeline.

The base neural network perception module is described in section 3.1 (page 4) and forms the baseline method without the proposed robustness modifications. This perception network ultimately learns latent state $s_t$ and features $f_t$ from which an action policy network $\pi(a_t^{\mathrm{net}}|f_t)$ is formed.

In section 3.2 (page 4), their formulism for a robust optimization planner is presented. They describe a notion of uncertainty through a set of object transition matrices $M^i$ (representing the i-th goal object's probability to move between cells each each $M^i_{uv}\in M^i$) on a discretized grid along with additional terms for prior $(\beta_i^t)_v$ and posterior $(\alpha_i^t)_v$ beliefs representing the agent's confidence that a target object occupies the cell. Using these terms, they seek to optimize the objective function of finding an agent path that maximizes the accumulated discounted capture beliefs (equation 4).

Section 3.3 (page 5), describes the usage of PICNNs to output a calibrated convex set of the prior discussed transition matrices and finally section 3.4 (page 6) describes how the resulting action $a_t^{\mathrm{task}}$ and reward $r_t^{\mathrm{task}}$ from the robust optimization procedure are used modulate the policy network and agent reward signal.

__Novelty and Contributions:__

The main novelty of the approach is to provide feedback from a robust optimization pipeline to a conventional neural network action policy. They do this by first using a Partially Input Convex Neural Network (PICNN) to determine a calibrated convex uncertainty set of object-tracking transition matrices from the base neural network's state vector. A robust optimization problem is solved using this set, and the resulting $a_t^\mathrm{task}$ and $r_t^\mathrm{task}$ are used to modulate the original action policies perceived reward and executed action (this overall process is described in section 3.4). They assert that this uncertainty-aware approach for tracking target objects during visual navigation is well suited for generalizing the the test environment under data scarce scenarios.

__Experiments:__

During experiments, the authors compare against and outperform a number of state of the art baselines (Table 1) in sequential and unordered multi-target object navigation (section 4.1). They hypothesize that the gain in formation is due to 1) the agent leveraging the additional reward and action signal from the downstream robust optimization solution and 2) the method’s ability to coordinate between multiple targets.

In section 4.2, the authors compare the NeuRO method vs the augmented neural network policy where their method obtains a significantly higher success rate versus training data (Figure 3).

Finally, experimental section 4.3 evaluates various ablation studies.

**Questions:**

- The detection factor ($d(p_t)_v$ in equation 3b) seems to be a relatively simple distance scale. I would of expected this value to be based on line of sight / obstacle occlusion as well. Can this be clarified? Is line of sight information incorporated in some other fashion when updating the target object belief state?

- In Table 3, could the authors please clarify what precisely "Inference time" refers to? On line 282 it states "inference time for solving the optimization problem..." Is this inference time only relevant during the training process to provide a reward feedback for training the agent's policy (as in Algorithm 1 on page 6) or is it also required during the testing phase? Currently this amount seems quite high for real-world usage if it is required at runtime.

- Given the method appears to be currently tied to a discrete grid for the transition matrices for representing object locations, do the authors have any thoughts on making use of this method in a continuous space and what the implications of that modification might be?

**Ethical Concerns:**

["NO or VERY MINOR ethics concerns only"]

**Final Justification:**

After reading reviewing the authors' rebuttal - and absent any further modifications - I have increased my overall score to `Accept`.

My primary original concerns were related to experimental rigour (missing experimental variance over repeated random seeds) and potential limited use (discrete world space, simple detection distance metric not accounting for sight information, inference times clarification).

Since then, the authors have addressed most of my stated weaknesses (including additional experimental results and clarifications). The issue of requiring a discrete world space still remains but is rationalized for practical efficiency reasons by the authors. Nonetheless, selection of the appropriate grid size without prior knowledge of the environment could pose an issue.

Please see my original review for a summary of the strengths and weaknesses of the work and author rebuttal comments for more details.

**Limitations:**

Yes. Limitations are listed by the authors on lines 329-335.

**Quality:**

3

**Strengths And Weaknesses:**

__Strengths:__

- __Novelty and performance (originality and significance):__ Novel approach which makes use of reward and action feedback from a downstream robust optimization process to modulate the agent's perceived rewards and action policy. This is an especially useful method for the case of data scarcity (such as for robotics). They note improved performance versus a wide array of existing state of the art baselines (Table 1). This process is further described above in the summary section.
- __Breadth of experiments (quality):__ This is done well for the most part (however, see weaknesses section below for additional comments). Comparison against range a range baselines is done (Table 1). Intuition regarding the improved results is provided (the agent leveraging the additional reward and action signal from the downstream robust optimization solution and 2) the method’s ability to coordinate between multiple targets). Ablation studies and hyper-parameter sweeps are also presented to provide justification regarding the selected method configuration. The authors further provide intuition as to the effects of these hyper parameters.
- __Manuscript clarity:__ The manuscript is generally well written and generally easy to understand.

__Weaknesses:__

- __Experimental rigour (quality):__ Results in experiments lack any standard deviation / confidence interval metrics for repeated trials (for example with different network random seeds). This would be especially important for some metrics which are quite close to baseline performance in Table 1 (PPL for S-mon (m=2) and Progress for S-Mon (m=3)).
- __Potential limited use (significance):__ The method appears to be limited to a discrete grid for object locations. In addition, the detection factor ($d(p_t)_v$ in equation 3b) seems to be a relatively simple distance scale. I would of expected this value to account for more information in the scene such as line of sight and obstacle occlusion as well. Likewise, the inference times in table 3 seem relatively large and require further clarification if this method can be used in real-time. See questions to below to authors for clarification.
- __Experimental description (clarity):__ There is a very limited amount of context when beginning to describe the experimental setup in section 4. Ideally I feel a brief introduction should be given and a reference to relevant sections in the appendix if needed.

__Overall Assessment and Suggestions for Improvement:__

Currently I have marked this work as borderline accept as it is novel and beats existing state of the art baselines (see strengths section above). However I feel that it could be improved with additional confidence interval analysis in the experiment section (see weaknesses above). There are also several limitations (noted in the weaknesses section) and the authors' own discussion regarding limitation to convex uncertainty sets (discussed on lines 329-335).

__Minor Points:__

- Lines 114-115 and 116-117 appear to repeat the same information.
- On line 163, I believe it should read “$i$ for conciseness”.
- Table 3 appears in the text before table 2. I imagine the label for these should be switched?

---

> ### Author Rebuttal · Authors · 2025-07-30
>
> We sincerely thank you for the constructive and detailed feedback. We are encouraged that you found our work novel, well-written, and outperforming SoTA baselines. We address the specific weaknesses (W1, W2, W3) and questions (Q1, Q2, Q3) below.
>
> ---
>
> ### **W1: Experimental Rigor**
>  We have now conducted experiments with 8 different random seeds and will update Table 1 with mean and standard deviation in the final version. (due to space limitations, we are currently able to include only a subset of the results.)
> |SMON$_{m=2}$: Test| | | | |
> |-|-|-|-|-|
> |**Methods**|**Success**|**Progress**|**SPL**|**PPL**|
> |OracleEgoMap|63.40 (±4.82)|71.92 (±2.18)|48.73 (±3.12)|56.12 (±1.62)|
> |Lyon|75.08 (±4.24)|83.44 (±1.81)|63.14 (±2.46)|68.33 (±1.73)|
> |NeuRO (Ours)|81.14 (±3.86)|85.26 (±1.53)|66.49 (±2.75)|71.10 (±1.42)|
>
> We observed that Progress and PPL provide inherently more robust counterparts to Success and SPL, respectively. Also, the new results show that NeuRO not only achieves a higher mean success rate but also exhibits significantly lower variance, further reinforcing our central claim regarding the model’s robustness and generalization capabilities.
>
> ---
> ### **W2: Potential Limited Use**
> Thank you for raising this point. We address it from three perspectives. Collectively, our analysis and experiments demonstrate that NeuRO is computationally efficient and effectively accounts for complex visual information in realistic environments, facilitating its applicability to real-time scenarios.
> 1. **Optimization under Continuous Uncertainty**: See **Q3**
> 2. **Incorporation of Sight Information**: See **Q1**.
> 3. **Inference Efficiency**: See **Q2**.
>
> ---
>
> ### **W3: Experimental Clarity**
> We thank you for pointing this out. Due to page limitations, we condensed the setup description. We will elaborate on these details in the revised version to improve clarity Specifically, we will:
> 1. Include a glossary and symbol table in the appendix, along with a description of the basic setup (e.g., action FORWARD (0.25m), TURN-LEFT/RIGHT (30°)) for the MultiON task.
> 2. Introduce prerequisite concepts such as conformal prediction at the beginning of the Method section.
> 3. Add more intuitive explanations to clarify key modeling choices (e.g., using a discrete grid for efficient optimization while maintaining continuous decision variables for differentiability).
>
> ---
> ### **Q1 & W2: Detection Factor & Sight Information**
> We appreciate the question and you are correct that $d(p_t)_v$ itself is a simplification. As for the line-of-sight information, we do explicitly incorporate it during the generation process of the uncertain transition matrix, where networks takes the raw depth image and BEV ego-map as inputs. We believe this incorporation contributes to our model’s strong performance even with a simple distance-based factor.
>
> To validate this hypothesis, we conducted a new experiment to make this geometric reasoning more explicit within the optimization process itself, inspired by your suggestion. We modified our PICNN to also predict an obstacle map (just like the current goal map) and merged two map to construct a optimization grid with both object location belief and obstacles. Then, we redefined $d(p_t)_v$ based on the geodesic distance (calculated via Breadth-First Search) instead original Manhattan one to consider the predicted obstacles. The results (**Progress** metric) are shown below:
> |Progress |SMON$_{m=2}$|SMON$_{m=3}$|UMON$_{m=2}$|UMON$_{m=3}$|
> |-|-|-|-|-|
> |NeuRO|85.26 (±1.53)|72.08 (±3.20)|64.33 (±3.82)|52.11 (±5.47)|
> |NeuRO + sight info.|87.90 (±1.26)|75.25 (±3.14)|67.26 (±3.35)|54.88 (±5.82)|
>
> As observed, this leads to a moderate but consistent improvement in navigation performance. We hypothesize this is because our perception module already captures parts of the occlusion information from the input depth data, so the contribution of the new explicit factor partially overlaps with what the network has already implicitly learned. This experiment affirms your intuition and further reinforces the physical grounding of our framework. We will include this insightful analysis in the revised version.
>
> ----
>
> ### **Q2 & W2: Inference Time**
> We thank you for this crucial clarifying question. The "Inference time" reported in Table 3 refers to the training-time process of solving the optimization problem and computing the policy gradients via Eq. 7(b), NOT the actual runtime deployment.
> 1. **Absolute Time**: The values in Table 3 are normalized for comparison (relative to $E=3, \tau=4$; see line 523-524 in Appendix). The absolute solving times are presented in Appendix F (Table 5) and are modest (e.g., 1.084s for a large $E=20, \tau=12$ setting). We will incorporate this clarification from the Appendix into the main paper.
> 2. **Runtime Performance**: We attribute the primary computational cost to the gradient computation of the optimization model but not the solving process, because modern linear program (LP) solvers are already highly efficient for solving. To validate this analysis, we measured the runtime cost at test stage, where the agent only needs to solve the forward pass of the optimization problem to get the action offset without any gradient computation. This result confirms that NeuRO is fully capable of real-time performance. We will add this crucial table and clarification to the revised version.
>     | |Memory Consumption |Test Runtime (SMON$_{m=2}$; per step (ms))|
>     |-|-|-|
>     |Lyon|22.18MB|12.9|
>     |HTP-GCN|25.63MB|15.3|
>     |NeuRO|20.24MB|11.6|
> 3. **Local Grid Scope**: The optimization time is manageable since our framework does not require a large optimization grid. This is because our optimization grid models the agent's local egocentric view, NOT the global map (Evidence: 1. The grid is re-formulated at every time step based on the current egocentric observation; 2. We always initialize agent position to the center of the optimization grid). This keeps the optimization problem small and tractable. In the MultiON task, the input ego-map spans a 5m view and each cell in the map represents a 0.8m*0.8m area, which means there are about 30 cells. Our generation process of the optimization grid can be understood as overlaying a goal position distribution onto this local map. Therefore, a grid size of $E=5$ ($V=25$ cells) is sufficient. In summary, our optimization grid simulates a local field of view, so the problem size remains bounded regardless of the overall environment scale.
> 4. **Scalability**: Furthermore, we have analyzed the source of the optimization problem’s computational complexity in Appendix F, which scales with $O(E^2\tau)$. To address this, we introduce a basis function expansion method in Appendix F to let network predict α and β. It effectively reduces this complexity to $O(K\tau)$, where $K \ll E^2$ is a constant. We empirically validate this approach, demonstrating its effectiveness in accelerating the optimization. However, for the sake of clarity and to maintain focus on the “network+optimization” framework itself in the main paper, we have detailed this extension in the appendix.
>
> ---
>
> ### **Q3 & W2: Continuous Space**
> This is a very insightful question that touches upon a fundamental challenge in robust optimization, but modifying the optimization grid (the uncertainty part in our robust optimization problem) to a continuous space would transform our problem into a semi-infinite program (SIP). In general, solving SIPs to optimality is NP-hard and computationally intractable, especially within the tight time constraints of an agent's decision loop. While relaxations can be used to approximate SIPs, they often alter the optimization landscape and sacrifice the guarantee of finding the true robust optimum.
>
> Therefore, we intentionally adopt a discretized grid as a principled and pragmatic design choice to ensure the downstream robust optimization problem remains a tractable Linear Program. This guarantees that we can find the global optimum of the subproblem efficiently and, more importantly, that we can differentiate efficiently through it for end-to-end training. Also, from a practical standpoint, one could even view the real world map as a very fine-grained discrete grid, making our approach a practical approximation. We will add this discussion to the appendix to provide a richer context for our design choices.

---

> > ### Comment · Reviewer_M9u1 · 2025-08-04
> >
> > Thank you for responding to my review and addressing my criticisms. I have raised my score to reflect this. I suppose my sole remaining concern is related to the requirement of a discrete world space. Though I appreciate your justification for practical computational efficiency.

---

> > > ### Author Response · Authors · 2025-08-05
> > >
> > > Thank you for your thoughtful response and for raising your score. We appreciate your engagement with our work and your constructive feedback. Regarding the discrete world space, we understand your concern and agree it’s an important modeling choice — we chose this design primarily to balance expressiveness and tractability, especially under real-time constraints. We hope future work can further explore this assumption.

---

### Decision · Program_Chairs · 2025-09-17

**Decision:**

Accept (poster)

**Comment:**

The submissions deals with visual navigation and data scarcity and proposes to add a task specific optimization routine, which deals with uncertainty in a dedicated way. All four reviews were quite appreciative and positive. The expert reviewers appreciated novelty (a fresh new perspective), sound methodology, the clear advantage of being able to handle uncertainty, performance on the Multi-ON task, comparisons with a large number of baselines, available code.

Some weaknesses have been pointed out: missing standard deviations, some limitations (grid; scalability; convexity), missing experiments with real robots, missing failure cases, and writing.

The rebuttal provided by the authors was helpful and could clear up many concerns.
There was a clear consensus on acceptance and the AC concurs.